# Locality-aware Gaussian Compression for Fast and High-quality Rendering

**Seungjoo Shin**[1]  **Jaesik Park**[2]  **Sunghyun Cho**[1]
[1]POSTECH  [2]Seoul National University
`https://seungjooshin.github.io/LocoGS`

## Abstract

We present LocoGS, a locality-aware 3D Gaussian Splatting (3DGS) framework that exploits the spatial coherence of 3D Gaussians for compact modeling of volumetric scenes. To this end, we first analyze the local coherence of 3D Gaussian attributes, and propose a novel locality-aware 3D Gaussian representation that effectively encodes locally-coherent Gaussian attributes using a neural field representation with a minimal storage requirement. On top of the novel representation, LocoGS is carefully designed with additional components such as dense initialization, an adaptive spherical harmonics bandwidth scheme and different encoding schemes for different Gaussian attributes to maximize compression performance. Experimental results demonstrate that our approach outperforms the rendering quality of existing compact Gaussian representations for representative real-world 3D datasets while achieving from $54.6\times$ to $96.6\times$ compressed storage size and from $2.1\times$ to $2.4\times$ rendering speed than 3DGS. Even our approach also demonstrates an averaged $2.4\times$ higher rendering speed than the state-of-the-art compression method with comparable compression performance.

## 1 Introduction

Advances in 3D representations have enabled the evolution of novel-view synthesis, which aims to model volumetric scenes to render photo-realistic novel-view images. Neural Radiance Field (NeRF) (Mildenhall et al., 2021), and its variants have demonstrated impressive performance in neural rendering from a set of RGB images and corresponding camera parameters. The advancements of radiance fields mainly focus on improving rendering quality (Zhang et al., 2020; Barron et al., 2021; 2022; 2023), accelerating convergence speed (Fridovich-Keil et al., 2022; Chen et al., 2022; Müller et al., 2022; Sun et al., 2022) and boosting rendering speed (Yu et al., 2021; Hedman et al., 2021; Chen et al., 2023; Reiser et al., 2023).

Recently, 3D Gaussian Splatting (3DGS) (Kerbl et al., 2023) proposes an efficient point-based 3D representation that represents volumetric scenes using explicit primitives known as 3D Gaussians. While existing radiance fields have encountered the trade-off between rendering quality and rendering speed for practical utilization, 3DGS has emerged as a promising 3D representation with notable advantages over previous methods, including high rendering quality and real-time rendering speed.

Despite its outstanding performance, 3DGS demands considerable storage costs due to the necessity to explicitly store numerous Gaussian parameters. In particular, representing a single 3D scene requires a large number of Gaussians, potentially millions, especially for complex scenes, while each Gaussian consists of several explicit attributes, including position, color, opacity, and covariance. As a result, the storage size can easily exceed 1GB. Therefore, there is a growing need for a compact 3DGS representation that requires a small storage size for a practical 3D representation.

To resolve the excessive storage requirement of 3DGS, several attempts have recently been made such as pruning techniques that prune less-contributing Gaussians (Lee et al., 2024; Fan et al., 2023; Girish et al., 2023). Quantization techniques have also been introduced to reduce storage requirements by representing Gaussian attributes through discretized data representations (Navaneet et al., 2024; Lee et al., 2024; Niedermayr et al., 2024; Girish et al., 2023; Fan et al., 2023). However, despite the remarkable compression performance, excessive quantization beyond a certain threshold significantly degrades rendering quality as their compression is performed solely based on the

global distributions of the attribute values without considering local contexts. Another noticeable line of work is anchor-based approaches (Lu et al., 2024; Chen et al., 2024). These approaches demonstrate an outstanding compression ratio and rendering quality by adopting view-adaptive neural Gaussian attributes, and a novel rendering scheme that decompresses neural Gaussian attributes and renders Gaussians together in a unified rendering pipeline. Despite their remarkable compression performance, their rendering scheme requires up to 30% additional rendering time compared to 3DGS, which makes them less practical for applications that require high-speed rendering, e.g., rendering of large-scale scenes on a mobile device.

To achieve high compression performance while maintaining the rendering efficiency of the 3DGS rendering pipeline, in this paper, we present *LocoGS*, a novel locality-aware compact 3DGS framework. As will be presented in the analysis in Section 4.1, Gaussians exhibit strong local coherence across nearby Gaussians. Inspired by this, our framework introduces a novel 3D Gaussian representation that leverages the spatial coherence of Gaussians. Our representation exploits a grid-based neural field representation that allows compact modeling of continuous physical quantities in volumetric scenes (Chen et al., 2022; Müller et al., 2022; Tang et al., 2022). Specifically, our representation encodes locally coherent Gaussian attributes into a multi-resolution hash grid (Müller et al., 2022) that enables a sparse voxel-grid representation to characterize the sparsely distributed structure of point-based representation. Unlike anchor-based approaches, our representation allows the usage of the original 3DGS rendering pipeline without any modification, thus maintaining the rendering efficiency of the original pipeline.

Besides the locality-aware 3D Gaussian representation, our framework also adopts several components to minimize the storage requirement and enhance the rendering speed and quality. Specifically, our framework adopts dense point cloud initialization, Gaussian pruning, an adaptive spherical harmonics (SH) bandwidth scheme, and quantization and encoding schemes specifically tailored for different Gaussian attributes. Combining all of these, our approach successfully outperforms the rendering quality of existing compact Gaussian representations while achieving from $54.6\times$ to $96.6\times$ compressed storage size and from $2.1\times$ to $2.4\times$ rendering speed than 3DGS. Our approach also demonstrates an averaged $2.4\times$ higher rendering speed than HAC (Chen et al., 2024), the state-of-the-art compression method, with comparable compression performance.

We summarize our contributions as follows:

- We introduce LocoGS, a locality-aware compact 3DGS framework that leverages the local coherence of 3D Gaussians to achieve a high compression ratio and rendering speed.

- To this end, we analyze the local coherence of Gaussians, which has been overlooked by previous work, and present a locality-aware 3D Gaussian representation.

- On top of the novel representation, LocoGS is carefully designed with additional components such as dense initialization, Gaussian pruning, an adaptive SH bandwidth scheme, and quantization and encoding schemes for different Gaussian attributes to maximize compression performance.

- Experimental results show that LocoGS clearly outperforms existing approaches in terms of compression performance and rendering speed.

## 2 RELATED WORK

**Neural Radiance Fields and Compact Representation**  Neural fields have emerged as a dominant representation for various continuous signals, including images (Dupont et al., 2021; Strümpler et al., 2022), signed distances (Yu et al., 2022; Wang et al., 2023), and 3D/4D scenes (Mildenhall et al., 2021; Barron et al., 2021; 2022; Müller et al., 2022). In particular, Neural Radiance Fields (NeRFs) (Mildenhall et al., 2021) have revolutionized the domain of neural rendering, exhibiting a remarkable ability to represent volumetric scenes. Recently, grid-based explicit representations have been adopted to encode local features and exploit them to model 3D scenes (Chen et al., 2022; Sun et al., 2022; Müller et al., 2022). Despite the outstanding performance, they yield limitations regarding the heavy storage burden. To address this issue, several approaches exploit efficient grid structure (Chen et al., 2022; Müller et al., 2022), pruning (Fridovich-Keil et al., 2022; Rho et al., 2023), and quantization (Takikawa et al., 2022; Shin & Park, 2024) to further consider efficiency in in terms of storage. Although compact neural radiance fields significantly reduce storage costs, the persistent issue of slow rendering speed remains a limiting factor to becoming a practical 3D representation. In this paper, we aim to accomplish real-time rendering via efficient point-based rendering while leveraging the compactness of neural fields to attain high rendering quality.

**3D Gaussian Splatting and Compact Representation**   While 3DGS (Kerbl et al., 2023) achieves real-time rendering and remarkable reconstruction performance, it requires excessive storage space due to the large quantity of 3D Gaussians needed for a single scene and the several explicit attributes necessary for each Gaussian. To address this issue, several 3D Gaussian compression approaches have been proposed, which can be categorized into global-statistic-based and locality-exploiting approaches. Global-statistic-based approaches do not leverage the local coherence of Gaussian attributes but rely on global statistics for quantization-based compression (Navaneet et al., 2024; Fan et al., 2023; Girish et al., 2023) and image-codec-based compression (Morgenstern et al., 2023). Locality-exploiting approaches include C3DGS (Niedermayr et al., 2024), Compact-3DGS (Lee et al., 2024), and anchor-based methods (Lu et al., 2024; Chen et al., 2024). These approaches exploit the local coherence of Gaussian attributes to reduce storage size. However, they still only partially exploit locality, resulting in limited performance compared to our method.

C3DGS (Niedermayr et al., 2024) sorts Gaussian attributes according to their positions along a space-filling curve, placing spatially close Gaussians at nearby positions in the sorted sequence. This enables subsequent compression steps to exploit local coherence. However, projecting 3D Gaussians to a 1D sequence along a space-filling curve cannot fully leverage the local coherence of Gaussian attributes. Compact-3DGS (Lee et al., 2024) uses a hash grid to encode view-dependent colors and vector quantization for geometric attributes such as rotation and scale. While it achieves a noticeable $81.8\times$ compression ratio for color, it only achieves a $5.5\times$ compression ratio for vector-quantized attributes, as it exploits global statistics instead of local coherence.

Anchor-based methods like Scaffold-GS (Lu et al., 2024) and HAC (Chen et al., 2024) also exploit local coherence. Scaffold-GS uses an anchor-point-based representation, where each anchor point encodes multiple locally-adjacent Gaussians with attributes changing based on the viewing direction. However, it focuses on rendering quality rather than compression and requires computationally expensive per-view processing. HAC builds on Scaffold-GS by incorporating a hash grid for compression, but unlike our method, it encodes only the context of anchor point features. Despite its high rendering quality and compression ratio, HAC requires slow per-view rendering due to its anchor-point-based representation.

Our approach also belongs to locality-exploiting approaches. In contrast to previous methods that partially exploit locality, our approach fully leverages the locality of all Gaussian attributes, including color and geometric attributes, for superior compression performance. To this end, we present an analysis of the local coherence of Gaussian attributes and propose a novel locality-based 3D Gaussian representation that directly encodes local information in an effective grid-based structure. Furthermore, our representation allows the use of the original 3DGS rendering pipeline, providing high rendering speed compared to anchor-based methods.

## 3   PRELIMINARIES: 3D GAUSSIAN SPLATTING

In this section, we briefly review 3DGS (Kerbl et al., 2023), which LocoGS is built upon. 3DGS represents a volumetric scene using a number of 3D Gaussians, each of which is a semi-transparent point-like primitive in a 3D space whose shape is defined by a 3D Gaussian function and whose color changes in a view-dependent way. Formally, a 3D Gaussian $\mathcal{G}$ can be defined as a set of its attributes, i.e., $\mathcal{G} = (\mathbf{p}, o, \mathbf{s}, \mathbf{r}, \mathbf{k})$, where $\mathbf{p} \in \mathbb{R}^3$ is a position, $o \in [0,1]$ is an opacity, $\mathbf{s} \in \mathbb{R}^3$ is a scale vector, and $\mathbf{r} \in \mathbb{R}^4$ is a quaternion-based rotation vector. $\mathbf{k} = (\mathbf{k}^0, \cdots, \mathbf{k}^L)$ is SH coefficients where $\mathbf{k}^l \in \mathbb{R}^{3 \times (2l+1)}$ is the SH coefficient of degree $l$, and $L$ is the maximum degree of the SH coefficients, which is commonly set to 3 to achieve sufficient quality (Kerbl et al., 2023). The opacity $\alpha$ of a 3D Gaussian $\mathcal{G}$ at a 3D coordinate $\mathbf{x} \in \mathbb{R}^3$ is defined as:

$$\alpha(\mathbf{x}; \mathcal{G}) = o \cdot \exp\left(-\frac{1}{2}(\mathbf{x} - \mathbf{p})^\top \Sigma^{-1}(\mathbf{x} - \mathbf{p})\right), \tag{1}$$

where $\Sigma = RSS^\top R^\top$ is a 3D covariance matrix, $R$ is a rotation matrix defined by $\mathbf{r}$, and $S$ is a scaling matrix defined as $S = \mathrm{diag}(\mathbf{s})$. Given a viewing direction $\mathbf{d}$, the view-dependent color of a 3D Gaussian $\mathcal{G}$ is computed as:

$$\mathbf{c}(\mathbf{d}; \mathcal{G}) = \mathbf{k}^0 + \sum_{l=1}^{L} H_l(\mathbf{d}; \mathbf{k}^l), \tag{2}$$

where $H_l$ is the SH function for the degree $l$.

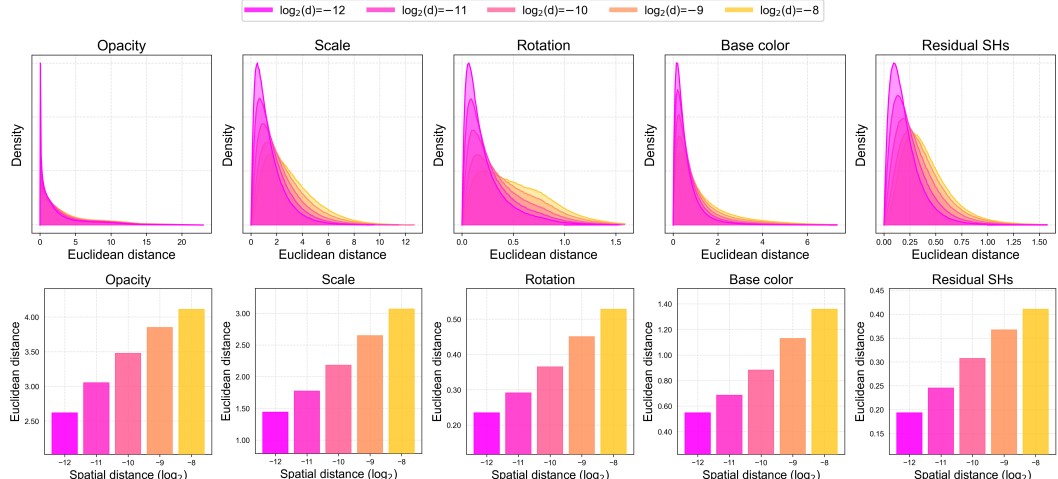

Figure 1: Evaluation of the local coherence of Gaussian attributes. We visualize histograms of the Euclidean distances of Gaussian attributes (top), and bar graphs of the average Euclidean distances of Gaussian attributes (bottom) between two Gaussians at different spatial distances. The yellow histograms and bar graphs correspond to the largest spatial distances, while the pink ones correspond to the smallest spatial distances.

The 3DGS method (Kerbl et al., 2023) reconstructs a target 3D scene from a collection of images by optimizing a set of 3D Gaussians. Once optimized, these 3D Gaussians can be projected to render novel-view images from any desired viewpoints. The optimization process begins with initializing the attributes of Gaussians employing sparse points from Structure-from-Motion (SfM) (Schonberger & Frahm, 2016). Precisely, the positions $\mathbf{p}$, scales $\mathbf{s}$, and base colors $\mathbf{k}^0$ are initialized based on the SfM-estimated points, while the other attributes are initialized as constants. The initialized attributes are then optimized to minimize the difference between the input images and the 2D renderings from the Gaussians. However, the initial number of Gaussians may not be sufficient to accurately represent the details in a complex real-world scene. Therefore, during the optimization process, additional Gaussians are introduced by cloning small Gaussians and splitting large Gaussians into smaller ones, thereby achieving a high-quality 3D reconstruction.

## 4 LOCALITY-AWARE 3D GAUSSIAN REPRESENTATION

### 4.1 ANALYSIS ON THE LOCAL COHERENCE OF 3D GAUSSIANS

Before introducing our locality-aware 3D Gaussian representation, we first analyze the local coherence of 3D Gaussians. For the analysis of the local coherence of Gaussians, we examine each Gaussian attribute reconstructed from widely used benchmark datasets. To this end, we apply 3DGS (Kerbl et al., 2023) to all the scenes from widely used benchmark datasets: Mip-NeRF 360 (Barron et al., 2022), Tanks and Temples (Knapitsch et al., 2017), and Deep Blending (Hedman et al., 2018), and obtained 3D Gaussians $\mathcal{G} = \{\mathcal{G}_i\}_{i=1}^N$. For each 3D Gaussian $\mathcal{G}_i$, we sample the neighboring Gaussians $\mathcal{G}_{\text{neighbor}}^i$ within a spatial distance $d \in \mathbb{R}^+$ as $\mathcal{G}_{\text{neighbor}}^i = \{\mathcal{G}_k\}_{k \in \mathcal{K}_i}$ where $\|\mathbf{p}_i - \mathbf{p}_k\|_2^2 < d$ and $\mathcal{K}_i$ denotes a set of Gaussian indices neighboring the $i$-th Gaussian $\mathcal{G}_i$. Specifically, we bound the position within $[0, 1]$ using the contraction function from Barron et al. (2022). Then, we compute the Euclidean distance of each attribute between $\mathcal{G}_i$ and its neighboring Gaussians $\mathcal{G}_{\text{neighbor}}^i$. Among them, we randomly sampled 100K pairs of Gaussians for each scene and computed their Euclidean distances.

Fig. 1 visualizes the Euclidean distances between two Gaussians with various spatial distances. It illustrates that, regardless of the spatial distances, all histograms have high peaks at small Euclidean distances, while their shapes become flatter as the spatial distances between Gaussians increase. The graphs demonstrate that spatially adjacent Gaussians have smaller Euclidean distances between their attributes. This local coherence among Gaussians is analogous to the pixel value coherence in natural images. Just as natural images display local color coherence, Gaussian attributes such as base color, SH coefficients, and opacity are often similar among adjacent Gaussians. Additionally,

the scale and rotation of a Gaussian are influenced by the local texture patterns and the underlying geometry, which are spatially coherent in natural images. Overall, our analysis demonstrates that Gaussian attributes possess significant local coherence.

## 4.2 Locality-aware 3D Gaussian Representation

Our main idea for a compact representation is to represent high-dimensional attributes, which are originally stored in an explicit form, using a neural field that supports a compact representation of spatially-coherent quantities. To this end, for a given 3D Gaussian primitive $\mathcal{G}$, we decompose its attributes into two groups, i.e., $\mathcal{G} = (\mathcal{G}^{\text{explicit}}, \mathcal{G}^{\text{implicit}})$ where $\mathcal{G}^{\text{explicit}}$ and $\mathcal{G}^{\text{implicit}}$ are the explicit attribute and implicit attribute, which are defined as:

$$\mathcal{G}^{\text{explicit}} = (\mathbf{p}, \gamma, \mathbf{k}^0), \qquad \text{and} \qquad \mathcal{G}^{\text{implicit}} = (o, \hat{\mathbf{s}}, \mathbf{r}, \mathbf{k}^{1:L}), \qquad (3)$$

respectively. In Eq. (3), $\gamma$ is a base scale attribute, and $\hat{\mathbf{s}}$ is a normalized scale attribute such that $\mathbf{s} = \gamma \hat{\mathbf{s}}$. $\mathbf{k}^0$ is the base color of Gaussian $\mathcal{G}$, which is the zero-frequency component of the SH coefficients, and $\mathbf{k}^{1:L}$ denotes the residual SH coefficients. Our representation stores $\mathcal{G}^{\text{explicit}}$ in an explicit form for each $\mathcal{G}$. On the other hand, $\mathcal{G}^{\text{implicit}}$ is stored in a compact neural field leveraging their local coherence to minimize the storage requirement. Specifically, we model $\mathcal{G}^{\text{implicit}}$ as $\mathcal{G}^{\text{implicit}} = \mathcal{F}(\mathbf{p})$ where $\mathcal{F}$ is a neural field so that we can retrieve the implicit attribute $\mathcal{G}^{\text{implicit}}$ for a given Gaussian primitive $\mathcal{G}$ using its position. For the neural field $\mathcal{F}$, we adopt the multi-resolution hash-grid approach (Müller et al., 2022). Specifically, $\mathcal{F}$ consists of a multi-resolution hash grid $\Phi_\theta$ followed by tiny multi-layer perceptrons (MLPs) $\Theta$, where $\theta$ is the parameter of $\Phi$.

We categorize the attributes into the explicit and implicit attributes to facilitate the 3D Gaussian optimization process while minimizing the storage requirement. As discussed in Section 3, the optimization process involves operations that directly manipulate the attribute values of Gaussian primitives such as initialization, cloning, and splitting, which are challenging with a neural field representation. Thus, $\mathcal{G}^{\text{explicit}}$ consists of the essential attributes that need to be stored in an explicit form to support direct manipulation despite that the explicit attributes also exhibit local coherence as shown in Section 4.1, while $\mathcal{G}^{\text{implicit}}$ consists of the remaining attributes for compact storage.

Fig. 2 illustrates an overview of our representation. Given a Gaussian primitive $\mathcal{G}$, we retrieve a local feature $\mathbf{f}$ corresponding to its implicit attribute by feeding its position $\mathbf{p}$, i.e., $\mathbf{f} = \texttt{interp}(\mathbf{p}, \Phi_\theta)$ where $\texttt{interp}$ is a tri-linear interpolation operator. The local feature $\mathbf{f}$ is then fed to the MLP of each of the attributes: normalized scale $\hat{\mathbf{s}}$, rotation $\mathbf{r}$, opacity $o$, and the residual SH coefficients $\mathbf{k}^{1:L}$. Finally, combining $\gamma$ with $\hat{\mathbf{s}}$ and $\mathbf{r}$, we obtain the covariance matrix of the Gaussian $\mathcal{G}$. Similarly, by concatenating $\mathbf{k}^0$ and $\mathbf{k}^{1:L}$, we obtain the full SH coefficients $\mathbf{k}$ of $\mathcal{G}$. To represent unbounded scenes using our representation, we adopt the coordinate contraction scheme of Barron et al. (2022) for the neural field representation $\mathcal{F}$.

**Adaptive SH Bandwidth** To further boost the rendering speed and quality of Gaussians, our framework adopts an adaptive SH bandwidth scheme inspired by Wang et al. (2024), where each Gaussian has a different number of SH coefficients. Specifically, for a Gaussian $\mathcal{G}$, its SH coefficient $\mathbf{k}$ is defined as $\mathbf{k} = \mathbf{k}^{0:b}$ where $b \in \{0, \cdots, L\}$ is an additional attribute of $\mathcal{G}$, which indicates the SH bandwidth, or the maximum degree of the SH coefficients. Based on the adaptive scheme, we redefine the explicit and implicit attributes in Eq. (3) as $\mathcal{G}^{\text{explicit}} = (\mathbf{p}, \gamma, \mathbf{k}^0, b)$ and $\mathcal{G}^{\text{implicit}} = (o, \hat{\mathbf{s}}, \mathbf{r}, \mathbf{k}^{1:b})$, respectively, where $b$ in $\mathcal{G}^{\text{explicit}}$ requires additional two bits for $L = 3$. While the adaptive SH scheme introduces a slight increase in the storage requirement, it greatly improves the rendering speed as a huge portion of surfaces in natural scenes are close to Lambertian surfaces whose colors do not change regardless of the view direction. Furthermore, it also improves the quality as it provides additional regularization on the colors of Gaussians.

## 5 LocoGS Framework

Fig. 3 illustrates the compression and decompression pipeline of LocoGS based on the locality-aware 3D Gaussian representation. Given a collection of input images of a target scene, our compression pipeline first performs the LocoGS representation learning step to learn a compact representation of the target scene, and further compresses the learned attributes through quantization and encoding. For decompression, our approach performs decoding and dequantization to reconstruct a LocoGS representation. Then, from the LocoGS representation, it reconstructs a conventional Gaussian representation that stores all the attributes in an explicit form for efficient rendering.

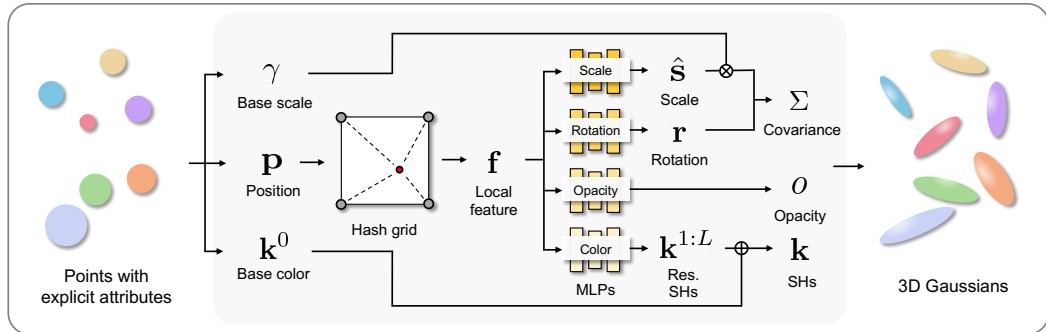

Figure 2: Overview of our framework: locality-aware 3D Gaussian representation.

Note that, in contrast to anchor-based methods (Lu et al., 2024; Chen et al., 2024), the decompression process of LocoGS reconstructs a conventional 3D Gaussian representation, which requires no alterations to the original 3DGS rendering pipeline that would compromise efficiency. Moreover, due to the high quality of our representation, we are able to more aggressively prune the less-contributing Gaussians while still preserving a rendering quality on par with prior methods, thereby enhancing the rendering speed. In the following, we describe each step of our compression pipeline in detail.

## 5.1 LOCALITY-AWARE 3D GAUSSIAN REPRESENTATION LEARNING

To build a compact 3D Gaussian representation, we follow the optimization process of 3DGS (Kerbl et al., 2023) with a slightly modified loss. Specifically, we optimize a loss $\mathcal{L}$ defined as:

$$\mathcal{L} = (1 - \lambda)\mathcal{L}_1 + \lambda\mathcal{L}_{\text{SSIM}} + \lambda_{\text{mask}}\mathcal{L}_{\text{mask}} + \lambda_{\text{mask}}^{\text{SH}}\mathcal{L}_{\text{mask}}^{\text{SH}}, \quad (4)$$

where $\mathcal{L}_1$ and $\mathcal{L}_{\text{SSIM}}$ represent an $L_1$ loss and an SSIM loss between images rendered from the compressed representation and their corresponding input images, respectively, both of which are borrowed from 3DGS. $\mathcal{L}_{\text{mask}}$ is a mask loss introduced by Lee et al. (2024) for pruning less-contributing Gaussians during the optimization. Also, $\mathcal{L}_{\text{mask}}^{\text{SH}}$ is a SH mask loss to restrict the maximum SH degree for each Gaussian. These mask losses will be further discussed later. $\lambda$, $\lambda_{\text{mask}}$, and $\lambda_{\text{mask}}^{\text{SH}}$ are balancing weights. We optimize the loss $\mathcal{L}$ with respect to the explicit attributes $\mathcal{G}^{\text{explicit}}$ of 3D Gaussians, and a hash grid $\Phi_\theta$ and MLPs $\Theta$ that encode implicit attributes $\mathcal{G}^{\text{implicit}}$. Additionally, we adopt mask parameters $\mu$ and $\boldsymbol{\eta}$ for each Gaussian to prune less-contributing Gaussians and select adaptive the SH bandwidths, optimizing $\mathcal{L}$ with respect to $\mu$ and $\boldsymbol{\eta}$ as well.

**Gaussian Pruning** The densification strategy of the 3DGS (Kerbl et al., 2023) optimization approach yields an excessive number of Gaussians, not only leading to a large storage size but also causing a degradation in rendering speed. To alleviate this, we adopt the soft movement pruning approach (Sanh et al., 2020; Lee et al., 2024). Specifically, we introduce an auxiliary mask parameter $\mu \in \mathbb{R}$ for each Gaussian, and apply it to the opacity and scale of the Gaussian as follows:

$$\tilde{o}_i = m_i o_i, \quad \tilde{\mathbf{s}}_i = m_i \mathbf{s}_i = m_i(\gamma_i \hat{\mathbf{s}}_i), \quad m_i = \mathbb{1}(\sigma(\mu_i) \geq \tau), \quad (5)$$

where the subscript $i$ is an index to each Gaussian. $\tilde{o}$ and $\tilde{\mathbf{s}}$ are the masked opacity and scale of the $i$-th Gaussian, respectively, which replace $o$ and $\mathbf{s}$ in the rendering process during optimization. $m_i \in \{0, 1\}$ is a binary mask computed from $\mu_i$ such that $m_i = 0$ means that the $i$-th Gaussian is pruned. $\mathbb{1}(\cdot)$ is an indicator function, $\sigma(\cdot)$ is a sigmoid function, and $\tau$ is a threshold value that controls the binary mask $m_i$. Since the indicator function is non-differentiable, we optimize the mask parameters $\mu_i$ employing the straight-through-estimator (STE) (Bengio et al., 2013). To control the sparsity level, the mask loss $\mathcal{L}_{\text{mask}}$ is defined as:

$$\mathcal{L}_{\text{mask}} = \frac{1}{N}\sum_{i=1}^{N}\sigma(\mu_i), \quad (6)$$

where $N$ indicates the number of Gaussians.

**Adaptive SH bandwidth** For adaptive SH bandwidth selection, we define an SH mask parameter $\boldsymbol{\eta} = (\eta^1, ..., \eta^L)^\top \in \mathbb{R}^L$ for each Gaussian and apply it to the residual SH coefficient $\mathbf{k}^{1:L}$ as:

$$\tilde{\mathbf{k}}_i^l = m_i^l \mathbf{k}_i^l, \quad m_i^l = \prod_{j=1}^{l}\mathbb{1}(\sigma(\eta_i^j) \geq \tau_{\text{SH}}), \quad (7)$$

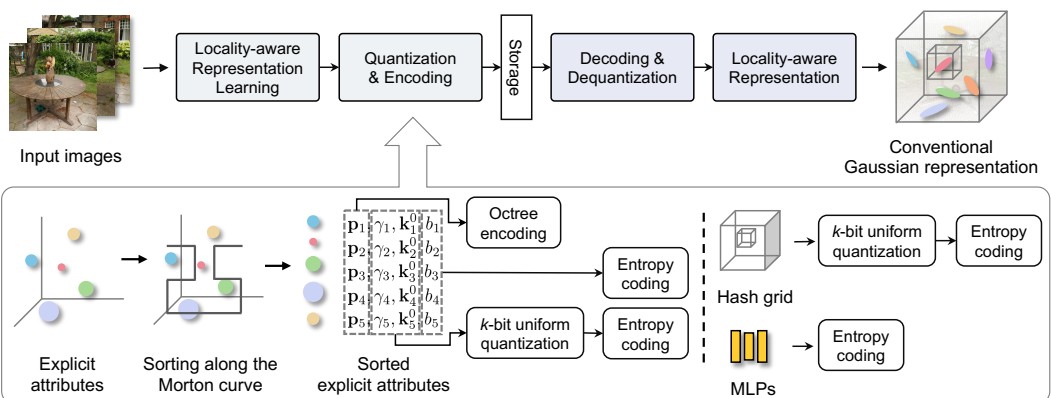

Figure 3: Overall pipeline of LocoGS.

where $i$ is an index to each Gaussian, $m_i^l \in \{0, 1\}$ is a binary mask to indicate whether the SH coefficients of the $l$-th degree are used, and $\tau_{\text{SH}}$ is a threshold value that controls the SH binary masks $m_i^l$. To control the SH bandwidth for each Gaussian, the SH mask loss $\mathcal{L}_{\text{mask}}^{\text{SH}}$ is defined as:

$$\mathcal{L}_{\text{mask}}^{\text{SH}} = \frac{1}{N} \sum_{i=1}^{N} \sum_{l=1}^{L} \frac{2l+1}{(L+1)^2 - 1} \sigma(\eta_i^l). \tag{8}$$

After optimization, we derive the adaptive SH bandwidth $b_i$ of the $i$-th Gaussian as:

$$b_i = \begin{cases} 0, & \text{if } m_i^1 = 0, \\ \max\{l | m_i^l = 1\}_{(1 \leq l \leq L)}, & \text{otherwise.} \end{cases} \tag{9}$$

**Initialization** The construction of a 3D Gaussian representation of a target scene requires an accurate initialization due to the highly non-linear nature of its optimization process. To initialize Gaussians, 3DGS (Kerbl et al., 2023) adopts a Structure-from-Motion (SfM) method (Schonberger & Frahm, 2016), known as COLMAP, that provides a sparse point cloud. However, a sparse point cloud is insufficient for capturing intricate details of complex real-world components. To mitigate this, we employ a dense point cloud generated from Nerfacto (Tancik et al., 2023). Specifically, we first build a neural representation for a given target scene using the approach of Nerfacto, and densely sample a point cloud adjacent to the reconstructed surface. From the neural representation, we sample one million points and initialize Gaussians using the sampled points. Refer to the `appendix` for more implementation details.

## 5.2 QUANTIZATION AND ENCODING

Once a 3D Gaussian representation of a target scene is obtained, we further compress both explicit and implicit attributes to achieve a more compact storage size. Fig. 3 shows an overview of the encoding process. To compress the explicit attributes, we employ different encoding schemes for different attributes. Specifically, we adopt G-PCC for the positions, which is an MPEG standard codec for geometry-based point cloud compression (Schwarz et al., 2018). For the other attributes, we adopt entropy coding. However, while the entropy coding scheme preserves the order of data, G-PCC does not, and consequently, loses the association between the positions and the other explicit attributes after encoding. Thus, to preserve the association between the positions and the other explicit attributes, we first sort the Gaussians along a Morton curve (Morton, 1966) before encoding the attributes. In the decompression stage, we reconstruct the association between the positions and the other attributes by sorting the positions after decoding them.

Regarding the positions, we adopt a lossless compression scheme. This is because the accuracy of 3D Gaussians is highly sensitive to positional errors, and even minor errors from lossy compression schemes like quantization can lead to significant quality degradation. To this end, we adopt G-PCC (Schwarz et al., 2018), which encodes 3D positions using an octree structure. While G-PCC significantly reduces storage requirements, its octree-based scheme requires positions quantized to

Table 1: Quantitative evaluation on Mip-NeRF 360 (Barron et al., 2022). We report the average scores of all scenes in the dataset. All storage sizes are in MB. We highlight the best score , second-best score , and third-best score of compact representations.

| Method | Mip-NeRF 360 | | | | |
|---|---|---|---|---|---|
| | PSNR ↑ | SSIM ↑ | LPIPS ↓ | Size ↓ | FPS ↑ |
| 3DGS (Kerbl et al., 2023) | 27.44 | 0.813 | 0.218 | 822.6 | 127 |
| Scaffold-GS (Lu et al., 2024) | 27.66 | 0.812 | 0.223 | 187.3 | 122 |
| CompGS (Navaneet et al., 2024) | 27.04 | 0.804 | 0.243 | 22.93 | 236 |
| Compact-3DGS (Lee et al., 2024) | 26.95 | 0.797 | 0.244 | 26.31 | 143 |
| C3DGS (Niedermayr et al., 2024) | 27.09 | 0.802 | 0.237 | 29.98 | 134 |
| LightGaussian (Fan et al., 2023) | 26.90 | 0.800 | 0.240 | 53.96 | 244 |
| EAGLES (Girish et al., 2023) | 27.10 | 0.807 | 0.234 | 59.49 | 155 |
| SSGS (Morgenstern et al., 2023) | 27.02 | 0.800 | 0.226 | 43.77 | 134 |
| HAC (Chen et al., 2024) | 27.49 | 0.807 | 0.236 | 16.95 | 110 |
| Ours-Small | 27.04 | 0.806 | 0.232 | 7.90 | 310 |
| Ours | 27.33 | 0.814 | 0.219 | 13.89 | 270 |

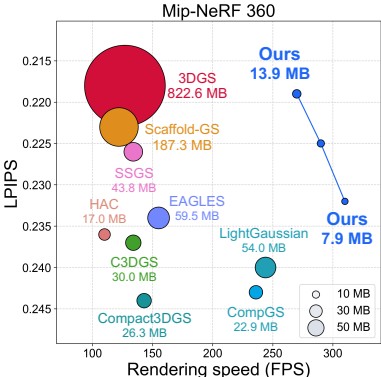

Figure 4: Comparison with baseline approaches on Mip-NeRF 360 (Barron et al., 2022).

the voxel resolution. To avoid information loss from quantization, we reinterpret the floating-point values of the positions as integer values with the same bit arrangement. This method prevents quantization errors and achieves lossless quantization of positions to integer values. We then encode these reinterpreted values using G-PCC. We refer the readers to `appendix` for more details on G-PCC.

For the base scale $\gamma$ and base color $\mathbf{k}^0$, we employ $k$-bit uniform quantization. Specifically, we quantize these attributes within the distributional range (Dupont et al., 2022), which is computed with the standard deviation of the parameters, rather than a simple Min-Max range. On the other hand, for the SH bandwidth $b$, we do not apply any quantization as it requires only two bits per Gaussian. Subsequently, we perform entropy coding on the quantized $\gamma$ and $\mathbf{k}^0$, and $b$. The entropy coding can be effectively performed thanks to the earlier sorting step. This step ensures that Gaussians, which are spatially proximate, are also closely placed in the sorted sequence. As a result, neighboring Gaussians in the sorted sequence demonstrate strong coherence, thereby boosting the efficiency of entropy encoding.

We also compress the implicit attributes, which are represented by the hash-grid $\Phi_\theta$ and following MLPs $\Theta$. Specifically, we apply both quantization and entropy coding to $\theta$. However, for $\Theta$, we only apply entropy coding as $\Theta$ is more susceptible to quantization errors.

## 6 EXPERIMENTS

**Implementation Details & Datasets**   Following the optimization scheme of 3DGS (Kerbl et al., 2023), the optimization process of LocoGS adopts 30K iterations to construct a 3D representation for each scene, which takes about one hour for a single scene on an NVIDIA RTX 6000 Ada GPU. During the optimization, we exponentially decay the learning rate for the neural field $\mathcal{F}$. We also employ 5K iterations of the warm-up stage to prevent the initial over-fitting of hash grid-based neural fields. Refer to `appendix` for more implementation details. For evaluation, we adopt three representative benchmark novel-view synthesis datasets. Specifically, we employ three real-world datasets: Mip-NeRF 360 (Barron et al., 2022), Tanks and Temples (Knapitsch et al., 2017), and Deep Blending (Hedman et al., 2018). Following 3DGS (Kerbl et al., 2023), we evaluate our method on nine scenes from Mip-NeRF 360, two from Tanks and Temples, and two from Deep Blending.

### 6.1 COMPARISON

We compare our method with existing compact 3D Gaussian frameworks as well as the original 3DGS (Kerbl et al., 2023) and Scaffold-GS (Lu et al., 2024). For comparison, we roughly categorize the existing compression approaches into the *quantization-based*, *image-codec-based*, *hash-grid-based*, and *anchor-based* approaches. The quantization-based approaches include CompGS (Navaneet et al., 2024), C3DGS (Niedermayr et al., 2024), LightGaussian (Fan et al., 2023), and EAGLES (Girish et al., 2023). The image-codec-based approach includes SSGS (Morgenstern et al., 2023), which maps 3D Gaussian attributes onto a 2D plane and compresses the attributes using an image codec. The hash-grid-based approaches include Compact-3DGS (Lee et al., 2024) as well as ours. Finally, the anchor-based approaches include Scaffold-GS (Lu et al., 2024) and HAC (Chen

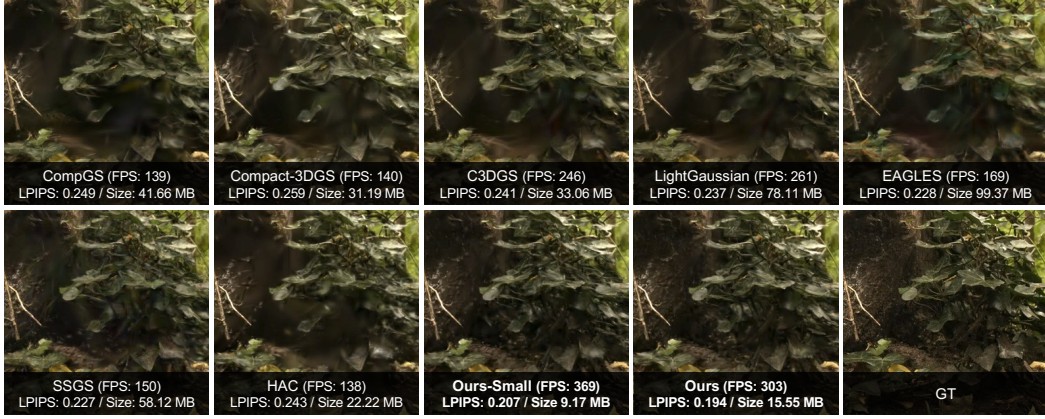

Figure 5: Qualitative results on the 'stump' scene of Mip-NeRF 360 (Barron et al., 2022). Rendering speed, LPIPS, and storage size are shown at the bottom of each subfigure.

Table 2: Quantitative evaluation on Tanks and Temples (Knapitsch et al., 2017) and Deep Blending (Hedman et al., 2018). We evaluate rendering quality, storage size, and rendering speed. We report the average scores of all scenes in the dataset. All storage sizes are in MB. We highlight the best score , second-best score , and third-best score of compact representations.

| Method | Tanks and Temples | | | | | Deep Blending | | | | |
|---|---|---|---|---|---|---|---|---|---|---|
| | PSNR ↑ | SSIM ↑ | LPIPS ↓ | Size ↓ | FPS ↑ | PSNR ↑ | SSIM ↑ | LPIPS ↓ | Size ↓ | FPS ↑ |
| 3DGS (Kerbl et al., 2023) | 23.67 | 0.844 | 0.179 | 452.4 | 175 | 29.48 | 0.900 | 0.246 | 692.5 | 134 |
| Scaffold-GS (Lu et al., 2024) | 24.11 | 0.855 | 0.165 | 154.3 | 109 | 30.28 | 0.907 | 0.243 | 121.2 | 194 |
| CompGS (Navaneet et al., 2024) | 23.29 | 0.835 | 0.201 | 14.23 | 329 | 29.89 | 0.907 | 0.253 | 15.15 | 301 |
| Compact-3DGS (Lee et al., 2024) | 23.33 | 0.831 | 0.202 | 18.97 | 199 | 29.71 | 0.901 | 0.257 | 21.75 | 184 |
| C3DGS (Niedermayr et al., 2024) | 23.52 | 0.837 | 0.188 | 18.58 | 166 | 29.53 | 0.899 | 0.254 | 24.96 | 143 |
| LightGaussian (Fan et al., 2023) | 23.32 | 0.829 | 0.204 | 29.94 | 379 | 29.12 | 0.895 | 0.262 | 45.25 | 287 |
| EAGLES (Girish et al., 2023) | 23.14 | 0.833 | 0.203 | 30.18 | 244 | 29.72 | 0.906 | 0.249 | 54.45 | 137 |
| SSGS (Morgenstern et al., 2023) | 23.54 | 0.833 | 0.188 | 24.42 | 222 | 29.21 | 0.891 | 0.271 | 19.32 | 224 |
| HAC (Chen et al., 2024) | 24.08 | 0.846 | 0.186 | 8.42 | 129 | 29.99 | 0.902 | 0.268 | 4.51 | 235 |
| Ours-Small | 23.63 | 0.847 | 0.169 | 6.59 | 333 | 30.06 | 0.904 | 0.249 | 7.64 | 334 |
| Ours | 23.84 | 0.852 | 0.161 | 12.34 | 311 | 30.11 | 0.906 | 0.243 | 13.38 | 297 |

et al., 2024), while Scaffold-GS adopts an anchor-based representation not for compression but for improving the rendering quality. For a fair comparison of rendering quality, storage size, and rendering speed, we reproduce the results of all the baseline models using the officially released codes and configurations. In our evaluation, we evaluate two variants of our approach: 'Ours', and 'Ours-Small'. 'Ours' is our baseline model, while 'Ours-Small' uses a smaller hash grid and more aggressive pruning compared to 'Ours' to obtain more compact representations.

Fig. 4, Table 1 and Table 2 show quantitative comparisons of the rendering quality (PSNR, SSIM, LPIPS), storage size, and rendering speed of different approaches. The comparison results show that our approach outperforms the baseline methods, achieving highly compact storage sizes (54.6× to 96.6× compressed storage size than 3DGS (Kerbl et al., 2023)), while achieving comparable or higher rendering quality. In terms of rendering speed, our approach offers 2.1× to 2.4× rendering speed compared to 3DGS. On the other hand, the quantization-based approaches (CompGS (Navaneet et al., 2024), C3DGS (Niedermayr et al., 2024), LightGaussian (Fan et al., 2023), and EAGLES (Girish et al., 2023)) suffer from relatively large storage sizes and lower rendering qualities as they solely rely on global statistics of Gaussian attributes not considering the local coherence. SSGS (Morgenstern et al., 2023), an image-codec-based approach, also exhibits lower compression performance as projecting 3D Gaussians to a 2D image plane cannot fully exploit the local coherence of 3D Gaussians. Compact-3DGS (Lee et al., 2024), which is a hash-grid-based approach, uses a hash grid only as a mapping to an RGB color from a 3D position and a view direction. Thus, it exhibits not only low compression ratios, but also slow rendering speed as its rendering process requires an additional MLP to obtain color values for a given view direction. Finally, HAC (Chen et al., 2024), an anchor-based approach, achieves impressive performance in terms of rendering quality and storage size, but its rendering speed is significantly slower than ours, as its rendering process also requires additional MLPs.

Table 3: Ablation study on the overall pipeline. All storage sizes are in MB.

| Method | Rendering | | | | | Storage size | | | | | |
|---|---|---|---|---|---|---|---|---|---|---|---|
| | PSNR ↑ | SSIM ↑ | LPIPS ↓ | FPS ↑ | #G | Position | Color | Scale | Mask | Hash+MLP | Total |
| 3DGS (Kerbl et al., 2023) | 27.44 | 0.813 | 0.218 | 127 | 3.32 M | - | - | - | - | - | 822.6 |
| 3DGS + Pruning + Compression | 26.95 | 0.798 | 0.235 | 188 | 1.54 M | - | - | - | - | - | 79.14 |
| Locality-aware 3D representation | 27.28 | 0.808 | 0.233 | 216 | 1.53 M | 8.17 | 8.17 | 2.72 | - | 25.26 | 44.32 |
| + Dense initialization | 27.36 | 0.814 | 0.219 | 238 | 1.32 M | 7.92 | 7.92 | 2.64 | - | 25.26 | 43.74 |
| + Adaptive SH bandwidth | 27.40 | 0.815 | 0.219 | 270 | 1.32 M | 7.95 | 7.95 | 2.65 | 0.33 | 25.26 | 44.14 |
| + G-PCC for positions | 27.40 | 0.815 | 0.219 | 270 | 1.32 M | 2.80 | 7.95 | 2.65 | 0.33 | 25.26 | 38.99 |
| + Explicit attr. compression | 27.35 | 0.814 | 0.219 | 270 | 1.32 M | 2.80 | 3.10 | 0.80 | 0.29 | 25.26 | 32.25 |
| + Implicit attr. compression | 27.33 | 0.814 | 0.219 | 270 | 1.32 M | 2.80 | 3.10 | 0.80 | 0.29 | 6.90 | 13.89 |

Fig. 5 demonstrates a qualitative comparison of LocoGS against previous methods (Navaneet et al., 2024; Lee et al., 2024; Niedermayr et al., 2024; Fan et al., 2023; Girish et al., 2023; Morgenstern et al., 2023; Chen et al., 2024). Existing compact approaches struggle with the intricate structures of real-world scenes, leading to severe artifacts (*e.g.*, floaters) due to the lack of information for achieving storage efficiency. In contrast, our method clearly shows superior rendering quality, capturing fine details of textureless regions, although it requires minimal storage capacity. Moreover, the result of 'Ours-Small', which requires a significantly smaller storage size, surpasses the rendering quality of all the other approaches.

## 6.2 ABLATION STUDY

**LocoGS Pipeline** To verify the effectiveness of each component in LocoGS, we conduct an ablation study. To this end, we evaluate the performance of variants of LocoGS on the Mip-NeRF 360 dataset (Barron et al., 2022). Table 3 reports the evaluation results. In the table, '3DGS + Pruning + Compression' indicates a variant of 3DGS, to which we apply Gaussian pruning and the compression schemes including G-PCC, quantization, and entropy coding as done in LocoGS without our locality-aware 3D representation. 'Locality-aware 3D representation' corresponds to our baseline model that adopts only the representation learning step and COLMAP for Gaussian initialization.

As the table shows, locality-aware representation greatly reduces the storage requirement compared to the '3DGS + Pruning + Compression', while maintaining a similar number of Gaussians. The dense initialization further enhances the rendering quality and speed while reducing the storage size as it helps accurately reconstruct the target scene with fewer Gaussians. The adaptive SH bandwidth scheme greatly enhances the rendering speed at the cost of a slightly increased storage size. Furthermore, G-PCC and the other compression steps for the explicit and implicit attributes introduce a significant storage size reduction while maintaining the rendering quality and speed. Firstly, G-PCC decreases 11.7% of storage space by compressing the positions. Moreover, explicit attribute compression saves 15.3% of storage capacity by compressing the base scales and base colors into smaller bits. Finally, implicit attribute compression achieves a further reduction of 41.6% in storage capacity by compressing the parameters of hash-grid and MLPs. Consequently, the compression pipeline shows a 3.18× compression ratio with a minimal loss in rendering performance. `Appendix` provides additional ablation studies on the dense initialization and adaptive SH bandwidth scheme.

## 7 CONCLUSION

In this paper, we introduced LocoGS, a locality-aware compact 3DGS framework that addresses large storage requirements of Gaussian primitives. To this end, we first explored the local coherence of Gaussian attributes and employed them to devise a storage-efficient representation. Based on the novel representation, we presented a carefully-designed framework with additional components to maximize compression performance. Our experiments verified that our approach demonstrates superior compression performance compared to existing approaches in terms of storage size, rendering quality, and rendering speed.

**Limitation** Despite its remarkable performance, our representation takes longer to train than existing methods, approximately one hour for each scene. In addition, our representation learning process requires more memory space than 3DGS due to the gradient computation of the hash grid and Gaussian attributes, which poses challenges for large-scale scenes. Thus, improving training efficiency is required by adopting optimization techniques to accelerate the computation of neural representations. Our method assumes a cube-shaped, object-centric scene for setting the hyperparameters of the hash grid, which may be less effective for complex structures in large-scale scenes.

## 8 ACKNOWLEDGMENTS

This work was supported by the National Research Foundation of Korea (NRF) grant funded by Korea government (Ministry of Education) (No.2022R1A6A1A03052954, Basic Science Research Program (10%)), and Institute of Information & communications Technology Planning & Evaluation (IITP) grant funded by Korea government (MSIT) (No.RS-2019-II191906, AI Graduate School Program (POSTECH) (10%); No.RS-2021-II211343, AI Graduate School Program (Seoul National University) (5%); No.RS-2024-00457882, AI Research Hub Project (30%); No.RS-2023-00227993, Detailed 3D reconstruction for urban areas from unstructured images (45%)).

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

# A APPENDIX

## A.1 IMPLEMENTATION AND EVALUATION DETAILS

**Architecture** The proposed neural fields consist of a multi-resolution hash grid with 16 levels of resolutions from 16 to 4096, followed by multi-layer perceptrons (MLPs) with two 64-channel hidden layers with ReLU activation for each locality-aware attribute. Each level of the hash grid contains up to $2^{17}$ feature vectors for our small variant and $2^{19}$ feature for our base model with two dimensions. The output of MLPs for each attribute is activated by the sigmoid function, normalization function, and exponential function for scale, rotation, and opacity, respectively. Note that no activation function is applied to spherical harmonics (SH) coefficients.

**Optimization** We optimize Nerfacto (Tancik et al., 2023) for 30K iterations to acquire a dense point cloud for initialization, which takes less than 10 min for a single scene. We set the rendering loss weight $\lambda = 0.2$, the mask threshold to $\tau = 0.01$ and the masking loss weight to $\lambda_{\text{mask}} = 0.005$ for our small variant and $\lambda_{\text{mask}} = 0.004$ for our base model. We set the SH mask threshold to $\tau_{\text{SH}} = 0.01$, and SH masking loss weight to $\lambda_{\text{mask}}^{\text{SH}} = 0.0001$.

**Quantization** We apply 6-bit uniform quantization for hash grid parameters $\theta$ and base scales $\gamma$ and apply 8-bit uniform quantization for base colors $\mathbf{k}^0$. Prior to $k$-bit quantization, we clip the target values to lie within $3 + 3(k-1)/15$ standard deviations of their mean, as proposed by Dupont et al. (2022). Additionally, we define position $\mathbf{p}$ with 16-bit precision with half tensor.

**Rendering Speed Measurement**   For a fair comparison of rendering speed, we reproduce all the baselines following the outlined configuration and measure the rendering speed in the same experimental setting. We render every test image 100 times on a single GPU (NVIDIA RTX 3090) and report the average rendering time. Also, we add 100 iterations of warm-up rendering before measuring the speed. We follow the original image resolution of the Tanks and Temples dataset (Knapitsch et al., 2017) and the Deep Blending dataset (Hedman et al., 2018) processed by 3DGS (Kerbl et al., 2023). In particular, we downsample the image resolution of the Mip-NeRF 360 dataset (Barron et al., 2022) not to exceed a maximum width of 1.6K, following the evaluation setting of 3DGS (Kerbl et al., 2023).

## A.2   ADDITIONAL DETAILS ON LocoGS

### A.2.1   DENSE INITIALIZATION

In the following, we introduce the detailed process to obtain a dense point cloud from optimized neural radiance fields (NeRF). NeRF optimizes a continuous 5D function that maps a 3D coordinate $\mathbf{x} \in \mathbb{R}^3$ and a 2D viewing direction $\mathbf{d} \in \mathbb{R}^2$ to an RGB color $\mathbf{c} \in \mathbb{R}^3$ and opacity $\sigma \in \mathbb{R}^+$. For volume rendering, the colors and opacities of sampled points along a ray $\mathbf{r}(t) = \mathbf{o} + t \cdot \mathbf{d}$ are accumulated to acquire the color of the ray as follows:

$$C(\mathbf{r}) = \sum_{i=1}^{N} T_i \alpha_i \mathbf{c}_i, \quad T_i = \prod_{j=1}^{i-1}(1 - \alpha_j), \quad \alpha_i = 1 - \exp(-\sigma_i \delta_i), \tag{10}$$

where $T_i$ denotes accumulated transmittance, $\alpha_i$ and $\mathbf{c}_i$ denote alpha and color of the $i$-th sampled point, respectively. $\delta_i = t_{i+1} - t_i$ denotes the distance between neighboring points. To acquire a dense point cloud, we render the median depth $z$ as follows:

$$z(\mathbf{r}) = \sum_{i=1}^{N_z} \alpha_i ||\mathbf{x}_i||, \quad \text{where} \quad T_{N_z} \geq 0.5 \quad \text{and} \quad T_{N_z-1} < 0.5, \tag{11}$$

where $\mathbf{x}_i$ is a sampled point along the ray and $N_z$ is the median index of sampled point along the ray $\mathbf{r}$. Then we project each training view pixel into 3D space according to its median depth $z$ as follows:

$$\mathcal{P}_{\text{dense}} = \{(\mathbf{p}_k, C(\mathbf{r}_k))\}_{k \in \mathcal{K}}, \quad \mathbf{p}_k = \mathbf{o}_k + z(\mathbf{r}_k)\mathbf{d}_k, \tag{12}$$

where $\mathbf{p}_k$ is the projected 3D coordinate along ray $\mathbf{r}_k$, $C(\mathbf{r}_k)$ denotes predicted color value along ray $\mathbf{r}_k$, and $\mathcal{K}$ is uniformly sampled ray indices for all rays of training views. $\mathbf{o}_k$ and $\mathbf{d}_k$ represents ray origin and ray direction, respectively. In this work, we set $|\mathcal{K}| = 1$M for efficiency.

### A.2.2   G-PCC

We explain the overall process of the octree-based geometry coding, G-PCC (Schwarz et al., 2018), for compressing positions. First, it voxelizes the input positions $\mathbf{p} \in \mathbb{R}^3$ before dividing octree nodes. This is achieved by computing quantized positions $\mathbf{p}' \in \mathbb{R}^3$ using user-defined scaling $s \in \mathbb{R}^+$ and shifting $\mathbf{p}_{\text{shift}} \in \mathbb{R}^3$ as follows:

$$\mathbf{p}' = \lfloor (\mathbf{p} - \mathbf{p}_{\text{shift}}) \times s \rfloor. \tag{13}$$

Upon these quantized positions, the octree structure is built through the recursive subdivision, and occupancy codes are efficiently stored using entropy coding. The resolution of the octree is determined by the scaling $s$, where a larger scaling requires more storage space. Conversely, using a smaller scaling reduces the storage size whereas it increases the quantization error in Eq. (13), resulting in accuracy reduction.

To analyze the impact of quantization error, we evaluate the rendering quality according to the varying quantization levels. Fig. 6 demonstrates that we encounter severe quality degradation above a specific quantization level, leading to limited compression performance. Thus, we propose a lossless transformation that ensures the high precision of positions for octree encoding, instead of using lossy quantization. The key idea is to reinterpret floating point values as integer values with the same bit arrangement as follows:

$$\mathbf{p}' = \text{sign}(\mathbf{p}) \times \text{float2int}(|\mathbf{p}|), \tag{14}$$

where $\text{float2int}()$ denotes a casting operator from float to integer. As shown in Fig. 6, the proposed invertible transformation enables lossless encoding and decoding of positions using octree geometry. Specifically, we convert `float16` into `int16`.

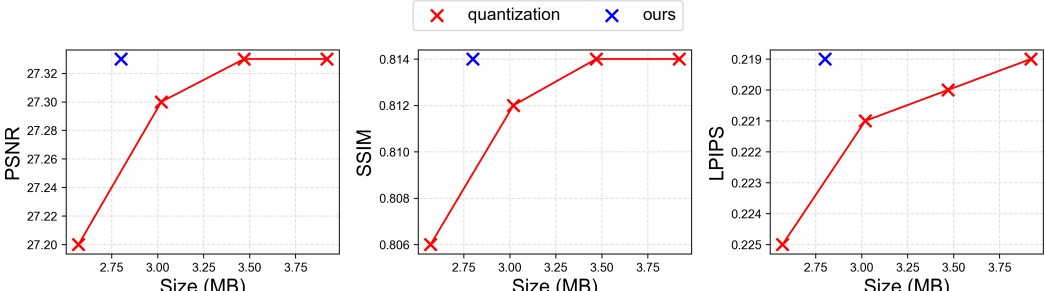

Figure 6: Ablation study on the voxelization method.

## A.3 ADDITIONAL ABLATION STUDIES

### A.3.1 ATTRIBUTE DESIGN

To validate the optimality of our current selection of explicit attributes, we compare the performance of LocoGS on the Mip-NeRF 360 dataset (Barron et al., 2022) when the base color and base scale are treated as either explicit or implicit attributes. Table 4 summarizes the comparison results. As the table shows, treating either the base scale or base color as implicit attributes degrades performance, as they cannot be explicitly initialized or manipulated during the training process. Specifically, treating the base scale as an implicit attribute makes the training process highly unstable, resulting in a severely small number of Gaussians and poor rendering quality. Treating the base color as an implicit attribute also degrades the rendering quality while increasing the number of Gaussians. Thus, we selected them as explicit attributes along with positions.

Table 4: Ablation study on the attribute design.

| Base color | Base scale | PSNR ↑ | SSIM ↑ | LPIPS ↓ | Size ↓ | FPS ↑ | #G |
|---|---|---|---|---|---|---|---|
| Explicit | Implicit | 20.48 | 0.661 | 0.383 | 10.13 | 176 | 0.65 M |
| Implicit | Explicit | 27.15 | 0.811 | 0.225 | 11.22 | 224 | 1.42 M |
| Explicit | Explicit | 27.33 | 0.814 | 0.219 | 13.89 | 270 | 1.32 M |

### A.3.2 HASH GRID SIZE

Our locality-aware 3D Gaussian representation significantly reduces storage by encoding Gaussian attributes in a compact hash grid. The size of the hash grid is a crucial parameter influencing both the compression ratio and quality. To understand the impact of the hash grid size, we examine the performance of our framework with different grid sizes using the Mip-NeRF 360 dataset (Barron et al., 2022). Specifically, we evaluate the performance according to the maximum number of feature vectors from $2^{17}$ to $2^{20}$, which are involved in each resolution level of the hash grid. Note that the hash grid size does not influence the rendering speed, as it does not affect the number of Gaussian primitives or spherical harmonics computation. As Fig. 7 demonstrates, a higher compression ratio can be achieved by reducing the hash grid size, although this leads to a decrease in rendering quality. The graphs also reveal that by increasing the hash grid size, we can attain a higher rendering quality than 3DGS (Kerbl et al., 2023), while the increased size is still about 40 times smaller than that of 3DGS. Furthermore, we can achieve a rendering quality comparable to Scaffold-GS (Lu et al., 2024) with just 10 MBs, which is more than 18 times smaller than that of Scaffold-GS.

### A.3.3 EFFECT OF DENSE INITIALIZATION ON LOCOGS

We analyze the effectiveness of the dense point cloud-based initialization. Fig. 8 demonstrates that the lack of geometric prior from a sparse point cloud results in visual artifacts in the rendering image, whereas a dense point cloud stands out for its ability to capture the fine details of the scene. As a result, Table 5 shows that we can further achieve robust optimization and photo-realistic rendering with dense geometric priors acquired from radiance fields.

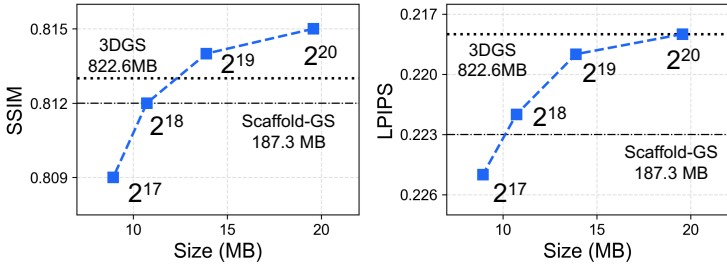

Figure 7: Ablation study on the hash grid size.

Table 5: Ablation study on the effect of dense initialization. 'Ours (w/o dense init.)' denotes our variant that begins optimization from SfM points, not applying dense initialization

| Method | PSNR ↑ | SSIM ↑ | LPIPS ↓ | Size ↓ | FPS ↑ |
|---|---|---|---|---|---|
| Ours (w/o dense init.) | 27.16 | 0.807 | 0.233 | 14.15 | 246 |
| Ours | 27.33 | 0.814 | 0.219 | 13.89 | 270 |

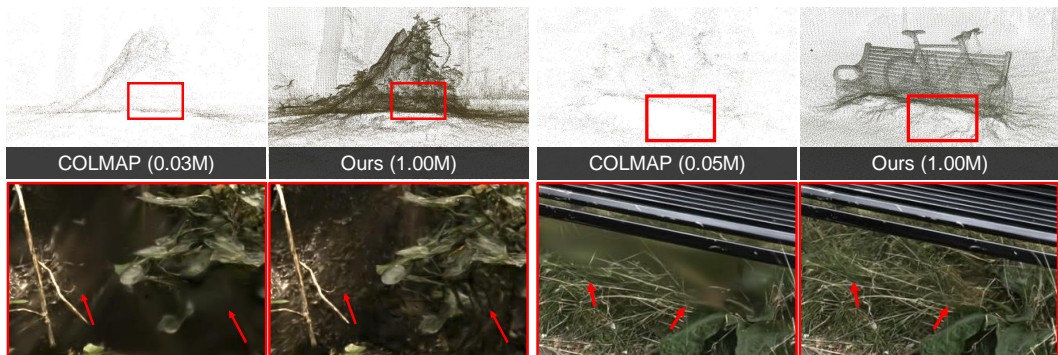

Figure 8: Visualization of the initial point cloud (upper) and rendering image (bottom).

### A.3.4 EFFECT OF DENSE INITIALIZATION ON EXISTING COMPRESSION METHODS

Moreover, we further adopt dense initialization in existing 3DGS compression methods to validate its general effectiveness. Specifically, we additionally adjust the compression parameters of the existing compression methods, such as pruning strength and quantization strength, since simply applying dense initialization increases the number of Gaussians. Thus, we conduct an evaluation comparing the performance of existing approaches a) without dense initialization, b) with dense initialization but no compression parameter tuning, and c) with dense initialization and compression parameter tuning on the MipNeRF-360 dataset (Barron et al., 2022). In this evaluation, we also include two non-compression methods, 3DGS (Kerbl et al., 2023) and Scaffold-GS (Lu et al., 2024), which have no compression parameters, so we do not perform compression parameter tuning for them. Fig. 9 and Table 6 summarize the evaluation results. As shown in Fig. 9(b), dense initialization without compression parameter tuning increases the number of Gaussians and storage size, and it decreases the rendering speed for all methods. As shown in Fig. 9(c), dense initialization with compression parameter tuning enhances rendering speed and storage requirements while achieving comparable or higher rendering quality for most existing approaches. It is worth mentioning that dense initialization with compression parameter tuning does not perform well with HAC (Chen et al., 2024), as shown in Table 6, because its anchor-based approach does not directly encode individual Gaussians, thus lacks a pruning mechanism to reduce the number of Gaussians. Compared to them, our approach still has distinctive advantages over existing compression methods producing smaller-sized results while providing higher rendering speed and comparable rendering quality.

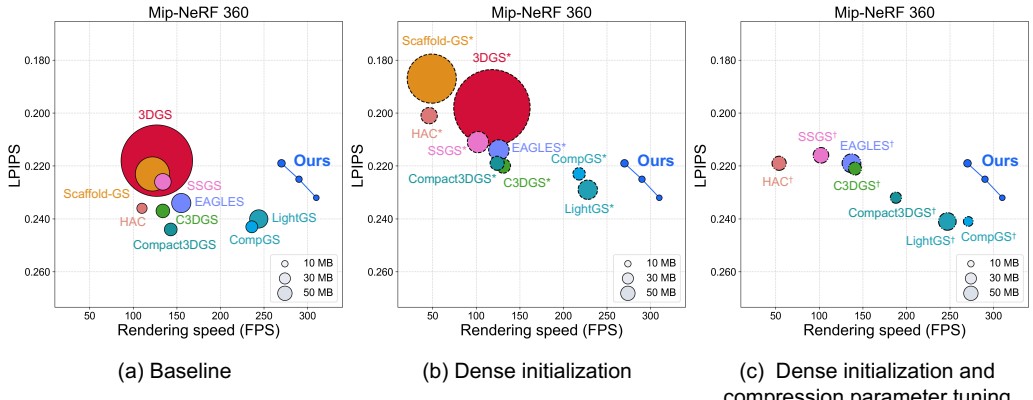

Figure 9: Ablation study on the dense initialization for existing 3DGS approaches. * denotes applying dense initialization, and † denotes applying dense initialization and compression parameter tuning.

Table 6: Ablation study on the dense initialization for existing 3DGS approaches. * denotes applying dense initialization, and † denotes applying dense initialization and compression parameter tuning.

| Method | PSNR ↑ | SSIM ↑ | LPIPS ↓ | Size ↓ | FPS ↑ | Method | PSNR ↑ | SSIM ↑ | LPIPS ↓ | Size ↓ | FPS ↑ |
|---|---|---|---|---|---|---|---|---|---|---|---|
| 3DGS | 27.44 | 0.813 | 0.218 | 822.6 | 127 | LightGaussian | 26.90 | 0.800 | 0.240 | 53.96 | 244 |
| 3DGS* | 27.80 | 0.826 | 0.198 | 935.8 | 118 | LightGaussian* | 27.08 | 0.806 | 0.229 | 60.21 | 228 |
| | | | | | | LightGaussian† | 26.86 | 0.799 | 0.241 | 49.64 | 247 |
| Scaffold-GS | 27.66 | 0.812 | 0.223 | 187.3 | 122 | EAGLES | 27.10 | 0.807 | 0.234 | 59.49 | 155 |
| Scaffold-GS* | 28.24 | 0.828 | 0.187 | 387.4 | 49 | EAGLES* | 27.49 | 0.821 | 0.214 | 67.54 | 126 |
| C3DGS | 27.09 | 0.802 | 0.237 | 29.98 | 134 | EAGLES† | 27.35 | 0.818 | 0.219 | 56.55 | 137 |
| C3DGS* | 27.39 | 0.812 | 0.220 | 33.87 | 131 | SSGS | 27.02 | 0.800 | 0.226 | 43.77 | 134 |
| C3DGS† | 27.39 | 0.812 | 0.221 | 26.95 | 141 | SSGS* | 27.13 | 0.805 | 0.211 | 72.00 | 102 |
| Compact-3DGS | 26.95 | 0.797 | 0.244 | 26.31 | 143 | SSGS† | 26.84 | 0.797 | 0.216 | 39.10 | 102 |
| Compact-3DGS* | 27.47 | 0.814 | 0.219 | 31.99 | 124 | HAC | 27.49 | 0.807 | 0.236 | 16.95 | 110 |
| Compact-3DGS† | 27.30 | 0.807 | 0.232 | 20.19 | 188 | HAC* | 27.71 | 0.822 | 0.201 | 42.67 | 46 |
| CompGS | 27.04 | 0.804 | 0.243 | 22.93 | 236 | HAC† | 27.16 | 0.808 | 0.219 | 32.19 | 54 |
| CompGS* | 27.42 | 0.818 | 0.223 | 24.75 | 218 | Ours | 27.33 | 0.814 | 0.219 | 13.89 | 270 |
| CompGS† | 27.21 | 0.810 | 0.241 | 15.24 | 271 | | | | | | |

### A.3.5 Dense Initialization Method

Table 7 compares the final rendering qualities of LocoGS initialized with 3DGS and Nerfacto. As shown in the table, LocoGS with Nerfacto achieves higher rendering quality, faster rendering speeds, and fewer Gaussians, thanks to the better initial solutions provided by Nerfacto. While Nerfacto typically produces results with lower rendering quality than 3DGS, it generates more accurate geometry information thanks to the regularization terms in its training loss function, which helps our training process achieve more accurate Gaussian representation learning. Furthermore, we find that Nerfacto also requires approximately $2\times$ shorter training time of 3DGS for both 30K iterations in the same environment (NVIDIA RTX 3090).

### A.3.6 Adaptive Spherical Harmonics Bandwidth

We investigate the effectiveness of the adaptive spherical harmonics (SH) bandwidth method compared to varying fixed SH bandwidth $L$. Table 8 implies that our SH pruning scheme adaptively reduces the SH bandwidth of less view-dependent Gaussians during optimization. Lower SH degrees yield notable rendering quality degradation despite the improvement in rendering speed. In contrast, we can accelerate the rendering process by excluding unnecessary SH elements, while the regularization effect of pruning leads to improved rendering quality compared to using the full SH degree ($L = 3$). Moreover, the results indicate that lower SH degrees ($L < 3$) tend to increase the number of Gaussians to represent the view-dependent effect through more Gaussians. As a result, our approach achieves faster speed due to less number of Gaussians, even compared to rendering without harmonics components ($L = 0$).

Table 7: Evaluation of LocoGS according to the dense initialization method.

| Method | PSNR ↑ | SSIM ↑ | LPIPS ↓ | Size ↓ | FPS ↑ | #G |
|--------|--------|--------|---------|--------|-------|-----|
| Bonsai (Mip-NeRF 360) | | | | | | |
| 3DGS | 31.55 | 0.933 | 0.212 | 9.39 | 315 | 0.52 M |
| Nerfacto | 31.58 | 0.935 | 0.208 | 9.15 | 355 | 0.49 M |
| Stump (Mip-NeRF 360) | | | | | | |
| 3DGS | 27.04 | 0.790 | 0.199 | 17.63 | 281 | 2.02 M |
| Nerfacto | 27.24 | 0.801 | 0.194 | 15.55 | 303 | 1.65 M |
| Truck (Tanks and Temples) | | | | | | |
| 3DGS | 25.34 | 0.880 | 0.129 | 13.17 | 303 | 1.07 M |
| Nerfacto | 25.70 | 0.885 | 0.125 | 12.19 | 326 | 0.90 M |
| Playroom (Deep Blending) | | | | | | |
| 3DGS | 30.32 | 0.907 | 0.250 | 11.55 | 328 | 0.83 M |
| Nerfacto | 30.49 | 0.907 | 0.248 | 10.75 | 386 | 0.68 M |

Table 8: Ablation study on the SH bandwidth.

| Method | PSNR ↑ | SSIM ↑ | LPIPS ↓ | Size ↓ | FPS ↑ |
|--------|--------|--------|---------|--------|-------|
| $L = 3$ | 27.29 | 0.814 | 0.219 | 13.58 | 238 |
| $L = 2$ | 27.22 | 0.813 | 0.220 | 13.64 | 258 |
| $L = 1$ | 27.08 | 0.811 | 0.221 | 13.68 | 268 |
| $L = 0$ | 26.72 | 0.804 | 0.226 | 13.96 | 269 |
| Ours | 27.33 | 0.814 | 0.219 | 13.89 | 270 |

## A.4 ALGORITHMS

We provide the locality-aware Gaussian representation learning step of LocoGS pipeline.

---

**Algorithm 1** Locality-aware Gaussian representation learning

---

$\mathcal{G}^{\text{explicit}}, \mathcal{F}, \mu, \boldsymbol{\eta} \leftarrow \text{INITIALIZE}()$      ▷ Explicit attribute, neural field, and mask parameter
**while** not converged **do**
     $\mathcal{G} \leftarrow \text{GETGAUSSIAN}(\mathcal{G}^{\text{explicit}}, \mathcal{F})$      ▷ Alg. 2
     $\tilde{\mathcal{G}} \leftarrow \text{MASKGAUSSIAN}(\mathcal{G}, \mu, \boldsymbol{\eta})$      ▷ Alg. 3
     $\mathcal{I} \leftarrow \text{RASTERIZE}(\tilde{\mathcal{G}})$      ▷ Rasterization
     $\mathcal{L} \leftarrow \text{LOSS}(\mathcal{I}, \mathcal{I}_{gt}) + \text{LOSS}(\mu) + \text{LOSS}(\boldsymbol{\eta})$      ▷ Eq. 4
     $\mathcal{G}^{\text{explicit}}, \mathcal{F}, \mu, \boldsymbol{\eta} \leftarrow \text{ADAM}(\nabla\mathcal{L})$
     **if** ISPRUNEITERATION($i$) **then**
         $\mathcal{G}^{\text{explicit}} \leftarrow \text{PRUNEGAUSSIAN}(\mathcal{G}^{\text{explicit}}, \mu)$      ▷ Alg. 4
     **end if**
     **if** ISREFINEMENTITERATION($i$) **then**
         $\mathcal{G}^{\text{explicit}} \leftarrow \text{ADAPTIVEDENSITYCONTROL}(\mathcal{G})$      ▷ Densification
     **end if**
**end while**
$b \leftarrow \text{COMPUTEBANDWIDTH}(\boldsymbol{\eta})$      ▷ Eq. 9

---

---

**Algorithm 2** Acquisition of Gaussians during optimization

---

**Input**: Explicit attributes $\mathcal{G}^{\text{explicit}} = \{\mathcal{G}_i^{\text{explicit}}\}_{i=1}^N$ and a neural field $\mathcal{F}$.
**Output**: Gaussian attributes $\mathcal{G}$
**function** GETGAUSSIAN($\mathcal{G}^{\text{explicit}}, \mathcal{F}$)
    $\mathcal{G}$ be a new list of Gaussians
    **for** $i \leftarrow 1$ to $N$ **do**                                                  $\triangleright$ $N$ Gaussians
        $\mathbf{p}_i, \gamma_i, \mathbf{k}_i^0 \leftarrow \mathcal{G}_i^{\text{explicit}}$                          $\triangleright$ Explicit attribute $\mathcal{G}_i^{\text{explicit}}$
        $o_i, \hat{\mathbf{s}}_i, \mathbf{r}_i, \mathbf{k}_i^{1:L} \leftarrow \mathcal{F}(\mathbf{p}_i)$                      $\triangleright$ Implicit attribute $\mathcal{G}_i^{\text{implicit}}$
        $\mathbf{s}_i \leftarrow \gamma_i \hat{\mathbf{s}}_i$                                    $\triangleright$ Scale $\mathbf{s}_i$
        $\mathbf{k}_i \leftarrow \{\mathbf{k}_i^0, \mathbf{k}_i^{1:L}\}$                         $\triangleright$ SH coefficients $\mathbf{k}_i$
        $\mathcal{G}_i \leftarrow \{\mathbf{p}_i, o_i, \mathbf{s}_i, \mathbf{r}_i, \mathbf{k}_i\}$                     $\triangleright$ Gaussian $\mathcal{G}_i$
    **end for**
    **return** $\mathcal{G}$                                        $\triangleright$ Gaussians $\mathcal{G}$
**end function**

---

**Algorithm 3** Masking Gaussians during optimization

---

**Input**: Gaussian attributes $\mathcal{G} = \{\mathcal{G}_i\}_{i=1}^N$ and corresponding mask parameters $\mu, \boldsymbol{\eta}$.
**Output**: Masked Gaussian attributes $\tilde{\mathcal{G}}$
**function** MASKGAUSSIAN($\mathcal{G}, \mu, \boldsymbol{\eta}$)
    $\tilde{\mathcal{G}}$ be a new list of masked Gaussians
    **for** $i \leftarrow 1$ to $N$ **do**                                               $\triangleright$ $N$ Gaussians
        $\mathbf{p}_i, o_i, \mathbf{s}_i, \mathbf{r}_i, \mathbf{k}_i \leftarrow \mathcal{G}_i$
        $m_i \leftarrow \mathbb{1}(\sigma(\mu_i) \leq \tau)$                              $\triangleright$ Binary mask $m_i$
        $\tilde{o}_i, \tilde{\mathbf{s}}_i \leftarrow m_i o, m_i \mathbf{s}_i$              $\triangleright$ Masked opacity $\tilde{o}_i$ and masked scale $\tilde{\mathbf{s}}_i$
        $m_i^0 \leftarrow 1$
        **for** $l \leftarrow 1$ to $L$ **do**
            $m_i^l \leftarrow m_i^{l-1} \cdot \mathbb{1}(\sigma(\eta_i^l) \leq \tau_{\text{SH}})$            $\triangleright$ Binary SH mask $m_i^l$ for degree $l$
            $\tilde{\mathbf{k}}_i^l \leftarrow m_i^l \mathbf{k}_i^l$                $\triangleright$ Masked SH coefficients $\tilde{\mathbf{k}}_i^l$ for degree $l$
        **end for**
        $\tilde{\mathcal{G}}_i \leftarrow \{\mathbf{p}_i, \tilde{o}_i, \tilde{\mathbf{s}}_i, \mathbf{r}_i, \tilde{\mathbf{k}}_i\}$                   $\triangleright$ Masked Gaussian $\tilde{\mathcal{G}}_i$
        **return** $\tilde{\mathcal{G}}_i$
    **end for**
    **return** $\tilde{\mathcal{G}}$                                 $\triangleright$ Masked Gaussians $\tilde{\mathcal{G}}$
**end function**

---

**Algorithm 4** Pruning Gaussian during optimization

---

**Input**: Explicit attributes $\mathcal{G}^{\text{explicit}} = \{\mathcal{G}_i^{\text{explicit}}\}_{i=1}^N$ and corresponding mask parameters $\mu$.
**Output**: Pruned explicit attributes $\mathcal{G}^{\text{explicit}}$
**function** PRUNEGAUSSIAN($\mathcal{G}^{\text{explicit}}, \mu$)
    **for** $i \leftarrow 1$ to $N$ **do**
        $m_i \leftarrow \mathbb{1}(\sigma(\mu_i) \leq \tau)$                              $\triangleright$ Binary mask $m_i$
        **if** $m_i = 0$ **then**
            $\mathcal{G}^{\text{explicit}} \leftarrow \mathcal{G}^{\text{explicit}} \backslash \mathcal{G}_i^{\text{explicit}}$               $\triangleright$ Remove $i$-th Gaussian
        **end if**
    **end for**
    **return** $\mathcal{G}^{\text{explicit}}$
**end function**

---

## A.5 ADDITIONAL RESULTS

### A.5.1 NUMBER OF GAUSSIANS

We evaluate the numbers of Gaussians after optimization in Table 9 and along the training iterations in Fig. 10. Compared to 'Ours', 'Ours-Small' has fewer Gaussians, resulting in higher rendering

speed and slight degradation in rendering quality. In our framework, the rendering speed is inversely proportional to the number of Gaussians. While the rendering quality is influenced by the number of Gaussians, as shown in Table 9, it is also affected by other factors, such as dense initialization. In Fig. 10, 'Ours' and 'Ours-Small' begin with dense initialization. Thus, at the beginning, they have more Gaussians than 3DGS and 'Ours (w/o dense init.)'. We prune Gaussians for the entire optimization process (30K iterations), whereas 3DGS preserves a large number of Gaussians after densification (15K iterations). Therefore, we can achieve a small Gaussian quantity after optimization. 'Ours-Small' uses a larger weight for the mask loss, so it has fewer Gaussians than 'Ours' after training.

Table 9: Evaluation of the rendering quality, rendering speed, and number of Gaussians.

| Method | Mip-NeRF 360 | | | Tanks and Temples | | | Deep Blending | | |
|---|---|---|---|---|---|---|---|---|---|
| | LPIPS ↓ | FPS ↑ | #G | LPIPS ↓ | FPS ↑ | #G | LPIPS ↓ | FPS ↑ | #G |
| Ours-Small | 0.232 | 310 | 1.09 M | 0.169 | 333 | 0.78 M | 0.249 | 334 | 1.04 M |
| Ours | 0.219 | 270 | 1.32 M | 0.161 | 311 | 0.89 M | 0.243 | 297 | 1.20 M |

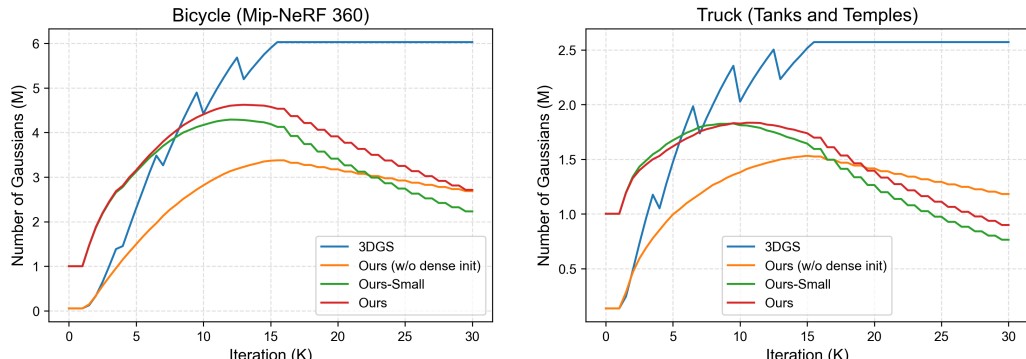

Figure 10: Illustration of the number of Gaussians along iterations. 'Ours (w/o dense init.)' denotes our variant that begins optimization from SfM points, not applying dense initialization.

### A.5.2 DECODING TIME

Table 10 compares the decoding times of different compression methods. LocoGS shows comparable decoding times despite its highly compact storage sizes compared to the other approaches, requiring less than nine seconds for a single scene.

Table 10: Evaluation of the decoding time (sec).

| Method | Mip-NeRF 360 | Tanks and Temples | Deep Blending |
|---|---|---|---|
| CompGS | 12.9 | 6.9 | 10.6 |
| Compact-3DGS | 178.4 | 139.4 | 59.8 |
| C3DGS | 0.2 | 0.1 | 0.2 |
| EAGLES | 8.1 | 4.1 | 7.0 |
| LightGaussian | 0.7 | 0.5 | 0.8 |
| SSGS | 6.6 | 3.8 | 2.8 |
| HAC | 24.3 | 13.9 | 5.0 |
| Ours-Small | 7.3 | 4.9 | 6.7 |
| Ours | 8.5 | 5.5 | 7.9 |

### A.5.3 LARGE-SCALE SCENES

We evaluate the performance of LocoGS on a representative large-scale dataset, the Mill-19 dataset (Turki et al., 2022), as shown in Table 11. The large-scale scenes in Mill-19 require an excessive number of Gaussians, resulting in large storage requirements exceeding 1.3GB for the

original 3DGS approach. LocoGS effectively reduces the memory size to less than 34MB, achieving a 47.2× compression ratio. Additionally, our effective compression scheme involving Gaussian point pruning also substantially improves the rendering speed.

Despite its high compression ratio, our method requires more memory space than the original 3DGS during training, which presents some challenges when applying our method directly to large-scale scenes. The slight quality degradation observed in Table 11 is also because of this heavy memory requirement, which limits the number of Gaussians to fewer than necessary. To address this issue, dividing a large-scale scene into smaller subregions could be a promising solution. Note that increased memory consumption only applies to the training phase. Our method follows memory usage of the original 3DGS for the rendering, which is proportional to the number of Gaussians.

Table 11: Evaluation on the Mill-19 dataset (Turki et al., 2022).

| Method | PSNR ↑ | SSIM ↑ | LPIPS ↓ | Size ↓ | FPS ↑ | #G |
|--------|--------|--------|---------|--------|-------|-----|
| Building | | | | | | |
| 3DGS | 19.14 | 0.651 | 0.365 | 1371.02 | 83 | 5.44 M |
| Ours | 19.79 | 0.630 | 0.354 | 32.97 | 113 | 4.13 M |
| Rubble | | | | | | |
| 3DGS | 23.75 | 0.733 | 0.311 | 1752.37 | 69 | 6.95 M |
| Ours | 23.38 | 0.651 | 0.366 | 33.26 | 122 | 4.20 M |

### A.5.4 QUANTITATIVE RESULTS

Tables 12, 13, and 14 demonstrate the quantitative results on each scene of the Mip-NeRF 360 dataset (Barron et al., 2022), Tanks and Temples dataset (Knapitsch et al., 2017), and Deep Blending dataset (Hedman et al., 2018). We measure the rendering quality (PSNR, SSIM, LPIPS), storage size, and rendering speed of each method on the various datasets.

### A.5.5 QUALITATIVE RESULTS

Figs. 11, 12, 13, 14, and 15 demonstrate the quantitative results on each scene of the Mip-NeRF 360 dataset (Barron et al., 2022), Tanks and Temples dataset (Knapitsch et al., 2017), and Deep Blending dataset (Hedman et al., 2018).

Table 12: Quantitative results of rendering quality, storage size, and rendering speed on Mip-NeRF 360 (Barron et al., 2022). All storage sizes are in MB. We highlight the best score, second-best score, and third-best score of compact representations.

| Method | Bicycle | | | | | Bonsai | | | | |
|---|---|---|---|---|---|---|---|---|---|---|
| | PSNR ↑ | SSIM ↑ | LPIPS ↓ | Size (MB) ↓ | FPS ↑ | PSNR ↑ | SSIM ↑ | LPIPS ↓ | Size (MB) ↓ | FPS ↑ |
| 3DGS (Kerbl et al., 2023) | 25.20 | 0.764 | 0.212 | 1506.3 | 69 | 32.13 | 0.940 | 0.206 | 310.7 | 208 |
| Scaffold-GS (Lu et al., 2024) | 25.21 | 0.758 | 0.231 | 309.8 | 101 | 32.52 | 0.941 | 0.205 | 130.7 | 117 |
| CompGS (Navaneet et al., 2024) | 24.98 | 0.750 | 0.238 | 49.15 | 99 | 31.48 | 0.932 | 0.216 | 13.74 | 171 |
| Compact-3DGS (Lee et al., 2024) | 24.81 | 0.739 | 0.256 | 38.55 | 90 | 31.80 | 0.931 | 0.219 | 14.91 | 210 |
| C3DGS (Niedermayr et al., 2024) | 24.91 | 0.752 | 0.248 | 35.64 | 191 | 31.18 | 0.930 | 0.226 | 10.41 | 288 |
| LightGaussian (Fan et al., 2023) | 24.93 | 0.752 | 0.233 | 98.82 | 181 | 30.84 | 0.926 | 0.225 | 20.55 | 317 |
| EAGLES (Girish et al., 2023) | 25.00 | 0.758 | 0.231 | 98.93 | 124 | 31.26 | 0.935 | 0.218 | 30.16 | 189 |
| SSGS (Morgenstern et al., 2023) | 24.64 | 0.745 | 0.218 | 56.93 | 131 | 31.39 | 0.930 | 0.207 | 25.40 | 136 |
| HAC (Chen et al., 2024) | 25.09 | 0.753 | 0.242 | 30.95 | 66 | 32.20 | 0.936 | 0.214 | 9.01 | 114 |
| Ours-Small | 25.32 | 0.760 | 0.227 | 13.75 | 141 | 31.18 | 0.928 | 0.220 | 4.11 | 401 |
| Ours | 25.41 | 0.770 | 0.211 | 21.28 | 135 | 31.58 | 0.935 | 0.208 | 9.15 | 355 |

| Method | Counter | | | | | Garden | | | | |
|---|---|---|---|---|---|---|---|---|---|---|
| | PSNR ↑ | SSIM ↑ | LPIPS ↓ | Size (MB) ↓ | FPS ↑ | PSNR ↑ | SSIM ↑ | LPIPS ↓ | Size (MB) ↓ | FPS ↑ |
| 3DGS (Kerbl et al., 2023) | 28.94 | 0.906 | 0.202 | 296.3 | 155 | 27.31 | 0.863 | 0.108 | 1419.7 | 79 |
| Scaffold-GS (Lu et al., 2024) | 29.38 | 0.908 | 0.203 | 89.5 | 115 | 27.32 | 0.858 | 0.119 | 227.3 | 99 |
| CompGS (Navaneet et al., 2024) | 28.61 | 0.896 | 0.215 | 14.16 | 138 | 26.91 | 0.847 | 0.138 | 48.56 | 107 |
| Compact-3DGS (Lee et al., 2024) | 28.54 | 0.892 | 0.223 | 14.00 | 153 | 26.61 | 0.838 | 0.145 | 36.52 | 100 |
| C3DGS (Niedermayr et al., 2024) | 28.33 | 0.893 | 0.226 | 10.69 | 217 | 26.90 | 0.848 | 0.142 | 37.26 | 201 |
| LightGaussian (Fan et al., 2023) | 27.97 | 0.885 | 0.232 | 19.53 | 234 | 26.78 | 0.844 | 0.136 | 92.81 | 189 |
| EAGLES (Girish et al., 2023) | 28.29 | 0.897 | 0.217 | 26.80 | 143 | 26.83 | 0.843 | 0.146 | 69.45 | 151 |
| SSGS (Morgenstern et al., 2023) | 28.56 | 0.894 | 0.210 | 19.00 | 139 | 26.98 | 0.847 | 0.119 | 88.68 | 101 |
| HAC (Chen et al., 2024) | 29.26 | 0.902 | 0.214 | 7.68 | 110 | 27.29 | 0.851 | 0.136 | 22.85 | 89 |
| Ours-Small | 28.40 | 0.889 | 0.224 | 4.42 | 238 | 26.81 | 0.834 | 0.155 | 7.26 | 349 |
| Ours | 28.72 | 0.897 | 0.211 | 9.56 | 214 | 27.07 | 0.845 | 0.139 | 13.36 | 302 |

| Method | Kitchen | | | | | Room | | | | |
|---|---|---|---|---|---|---|---|---|---|---|
| | PSNR ↑ | SSIM ↑ | LPIPS ↓ | Size (MB) ↓ | FPS ↑ | PSNR ↑ | SSIM ↑ | LPIPS ↓ | Size (MB) ↓ | FPS ↑ |
| 3DGS (Kerbl et al., 2023) | 31.53 | 0.926 | 0.127 | 445.1 | 123 | 31.38 | 0.917 | 0.221 | 379.6 | 144 |
| Scaffold-GS (Lu et al., 2024) | 31.48 | 0.925 | 0.129 | 106.4 | 90 | 32.04 | 0.922 | 0.213 | 87.7 | 154 |
| CompGS (Navaneet et al., 2024) | 30.83 | 0.916 | 0.138 | 19.12 | 121 | 31.11 | 0.911 | 0.231 | 15.44 | 138 |
| Compact-3DGS (Lee et al., 2024) | 30.38 | 0.913 | 0.141 | 22.11 | 119 | 30.69 | 0.910 | 0.233 | 13.92 | 199 |
| C3DGS (Niedermayr et al., 2024) | 30.47 | 0.916 | 0.144 | 15.45 | 200 | 31.13 | 0.911 | 0.238 | 9.19 | 256 |
| LightGaussian (Fan et al., 2023) | 30.36 | 0.908 | 0.152 | 29.16 | 221 | 30.95 | 0.908 | 0.238 | 24.95 | 235 |
| EAGLES (Girish et al., 2023) | 30.56 | 0.921 | 0.138 | 47.35 | 115 | 31.44 | 0.917 | 0.226 | 31.57 | 124 |
| SSGS (Morgenstern et al., 2023) | 30.59 | 0.907 | 0.147 | 24.73 | 121 | 30.79 | 0.905 | 0.229 | 15.32 | 133 |
| HAC (Chen et al., 2024) | 30.99 | 0.917 | 0.141 | 8.59 | 87 | 31.38 | 0.912 | 0.233 | 5.63 | 155 |
| Ours-Small | 30.18 | 0.907 | 0.156 | 4.41 | 349 | 31.18 | 0.905 | 0.249 | 3.84 | 360 |
| Ours | 30.71 | 0.914 | 0.143 | 9.47 | 298 | 31.44 | 0.913 | 0.232 | 9.19 | 318 |

| Method | Stump | | | | | Flowers | | | | |
|---|---|---|---|---|---|---|---|---|---|---|
| | PSNR ↑ | SSIM ↑ | LPIPS ↓ | Size (MB) ↓ | FPS ↑ | PSNR ↑ | SSIM ↑ | LPIPS ↓ | Size (MB) ↓ | FPS ↑ |
| 3DGS (Kerbl et al., 2023) | 26.59 | 0.770 | 0.217 | 1193.8 | 110 | 21.44 | 0.602 | 0.340 | 906.3 | 134 |
| Scaffold-GS (Lu et al., 2024) | 26.56 | 0.766 | 0.235 | 259.1 | 147 | 21.36 | 0.593 | 0.349 | 230.0 | 142 |
| CompGS (Navaneet et al., 2024) | 26.32 | 0.757 | 0.249 | 41.66 | 139 | 21.16 | 0.584 | 0.358 | 33.02 | 154 |
| Compact-3DGS (Lee et al., 2024) | 26.14 | 0.751 | 0.259 | 31.19 | 140 | 21.00 | 0.568 | 0.378 | 28.60 | 159 |
| C3DGS (Niedermayr et al., 2024) | 26.52 | 0.767 | 0.241 | 33.06 | 246 | 21.29 | 0.587 | 0.368 | 27.83 | 274 |
| LightGaussian (Fan et al., 2023) | 26.46 | 0.765 | 0.237 | 78.11 | 261 | 21.31 | 0.587 | 0.358 | 59.61 | 304 |
| EAGLES (Girish et al., 2023) | 26.59 | 0.771 | 0.228 | 99.37 | 169 | 21.28 | 0.588 | 0.361 | 58.04 | 219 |
| SSGS (Morgenstern et al., 2023) | 26.10 | 0.749 | 0.227 | 58.12 | 150 | 21.62 | 0.604 | 0.332 | 59.02 | 145 |
| HAC (Chen et al., 2024) | 26.62 | 0.765 | 0.243 | 22.22 | 138 | 21.39 | 0.587 | 0.360 | 21.65 | 127 |
| Ours-Small | 27.04 | 0.792 | 0.207 | 9.17 | 369 | 21.55 | 0.613 | 0.317 | 10.46 | 315 |
| Ours | 27.24 | 0.801 | 0.194 | 15.55 | 303 | 21.59 | 0.623 | 0.306 | 17.14 | 268 |

| Method | Treehill | | | | | | | | | |
|---|---|---|---|---|---|---|---|---|---|---|
| | PSNR ↑ | SSIM ↑ | LPIPS ↓ | Size (MB) ↓ | FPS ↑ | | | | | |
| 3DGS (Kerbl et al., 2023) | 22.48 | 0.633 | 0.327 | 945.8 | 120 | | | | | |
| Scaffold-GS (Lu et al., 2024) | 23.02 | 0.642 | 0.324 | 245.5 | 130 | | | | | |
| CompGS (Navaneet et al., 2024) | 22.41 | 0.622 | 0.351 | 34.99 | 142 | | | | | |
| Compact-3DGS (Lee et al., 2024) | 22.61 | 0.634 | 0.347 | 37.00 | 116 | | | | | |
| C3DGS (Niedermayr et al., 2024) | 22.66 | 0.632 | 0.359 | 26.85 | 252 | | | | | |
| LightGaussian (Fan et al., 2023) | 22.53 | 0.625 | 0.350 | 62.05 | 251 | | | | | |
| EAGLES (Girish et al., 2023) | 22.62 | 0.637 | 0.338 | 73.73 | 161 | | | | | |
| SSGS (Morgenstern et al., 2023) | 22.52 | 0.615 | 0.348 | 46.69 | 149 | | | | | |
| HAC (Chen et al., 2024) | 23.22 | 0.642 | 0.341 | 23.95 | 103 | | | | | |
| Ours-Small | 22.80 | 0.632 | 0.334 | 13.65 | 267 | | | | | |
| Ours | 22.86 | 0.635 | 0.328 | 20.33 | 237 | | | | | |

Table 13: Quantitative results of rendering quality, storage size, and rendering speed on Tanks and Temples (Knapitsch et al., 2017). All storage sizes are in MB. We highlight the ◼best score◼, ◼second-best score◼, and ◼third-best score◼ of compact representations.

| Method | Train | | | | | Truck | | | | |
|---|---|---|---|---|---|---|---|---|---|---|
| | PSNR ↑ | SSIM ↑ | LPIPS ↓ | Size ↓ | FPS ↑ | PSNR ↑ | SSIM ↑ | LPIPS ↓ | Size ↓ | FPS ↑ |
| 3DGS (Kerbl et al., 2023) | 21.95 | 0.810 | 0.210 | 266.6 | 201 | 25.38 | 0.878 | 0.148 | 638.2 | 150 |
| Scaffold-GS (Lu et al., 2024) | 22.50 | 0.828 | 0.190 | 110.5 | 112 | 25.71 | 0.882 | 0.139 | 198.1 | 106 |
| CompGS (Navaneet et al., 2024) | 21.83 | 0.802 | 0.221 | 14.40 | 170 | 25.22 | 0.872 | 0.156 | 22.75 | 162 |
| Compact-3DGS (Lee et al., 2024) | 21.64 | 0.792 | 0.241 | 17.64 | 203 | 25.02 | 0.870 | 0.163 | 20.30 | 196 |
| C3DGS (Niedermayr et al., 2024) | 21.61 | 0.800 | 0.233 | 13.61 | 311 | 24.98 | 0.871 | 0.168 | 14.85 | 346 |
| LightGaussian (Fan et al., 2023) | 21.54 | 0.788 | 0.248 | 17.75 | 440 | 25.10 | 0.870 | 0.161 | 42.14 | 318 |
| EAGLES (Girish et al., 2023) | 21.28 | 0.794 | 0.240 | 21.49 | 247 | 25.00 | 0.871 | 0.165 | 38.87 | 242 |
| SSGS (Morgenstern et al., 2023) | 21.64 | 0.794 | 0.225 | 17.19 | 254 | 25.45 | 0.872 | 0.150 | 31.64 | 190 |
| HAC (Chen et al., 2024) | 22.22 | 0.813 | 0.217 | 7.10 | 136 | 25.95 | 0.879 | 0.155 | 9.75 | 122 |
| Ours-Small | 21.71 | 0.814 | 0.203 | 6.75 | 311 | 25.53 | 0.880 | 0.134 | 6.43 | 356 |
| Ours | 22.07 | 0.822 | 0.195 | 12.49 | 296 | 25.70 | 0.885 | 0.125 | 12.19 | 326 |

Table 14: Quantitative results of rendering quality, storage size, and rendering speed on Deep Blending (Hedman et al., 2018). All storag sizes are in MB. We highlight the ◼best score◼, ◼second-best score◼, and ◼third-best score◼ of compact representations.

| Method | Drjohnson | | | | | Playroom | | | | |
|---|---|---|---|---|---|---|---|---|---|---|
| | PSNR ↑ | SSIM ↑ | LPIPS ↓ | Size ↓ | FPS ↑ | PSNR ↑ | SSIM ↑ | LPIPS ↓ | Size ↓ | FPS ↑ |
| 3DGS (Kerbl et al., 2023) | 29.14 | 0.898 | 0.246 | 810.9 | 110 | 29.83 | 0.901 | 0.247 | 574.2 | 157 |
| Scaffold-GS (Lu et al., 2024) | 29.84 | 0.906 | 0.242 | 137.9 | 167 | 30.72 | 0.908 | 0.245 | 104.4 | 221 |
| CompGS (Navaneet et al., 2024) | 29.14 | 0.897 | 0.254 | 28.97 | 128 | 29.91 | 0.900 | 0.254 | 20.94 | 157 |
| Compact-3DGS (Lee et al., 2024) | 29.16 | 0.899 | 0.257 | 25.97 | 152 | 30.26 | 0.902 | 0.258 | 17.53 | 216 |
| C3DGS (Niedermayr et al., 2024) | 29.46 | 0.906 | 0.251 | 19.47 | 251 | 30.32 | 0.907 | 0.255 | 10.83 | 351 |
| LightGaussian (Fan et al., 2023) | 28.75 | 0.894 | 0.262 | 52.97 | 247 | 29.49 | 0.896 | 0.261 | 37.54 | 328 |
| EAGLES (Girish et al., 2023) | 29.32 | 0.906 | 0.244 | 71.52 | 98 | 30.12 | 0.907 | 0.254 | 37.38 | 176 |
| SSGS (Morgenstern et al., 2023) | 28.81 | 0.886 | 0.277 | 19.98 | 210 | 29.61 | 0.897 | 0.264 | 18.66 | 239 |
| HAC (Chen et al., 2024) | 29.56 | 0.903 | 0.265 | 5.59 | 196 | 30.43 | 0.902 | 0.272 | 3.44 | 275 |
| Ours-Small | 29.85 | 0.904 | 0.244 | 10.22 | 214 | 30.40 | 0.906 | 0.256 | 5.06 | 454 |
| Ours | 29.84 | 0.904 | 0.240 | 16.02 | 208 | 30.49 | 0.907 | 0.248 | 10.75 | 386 |

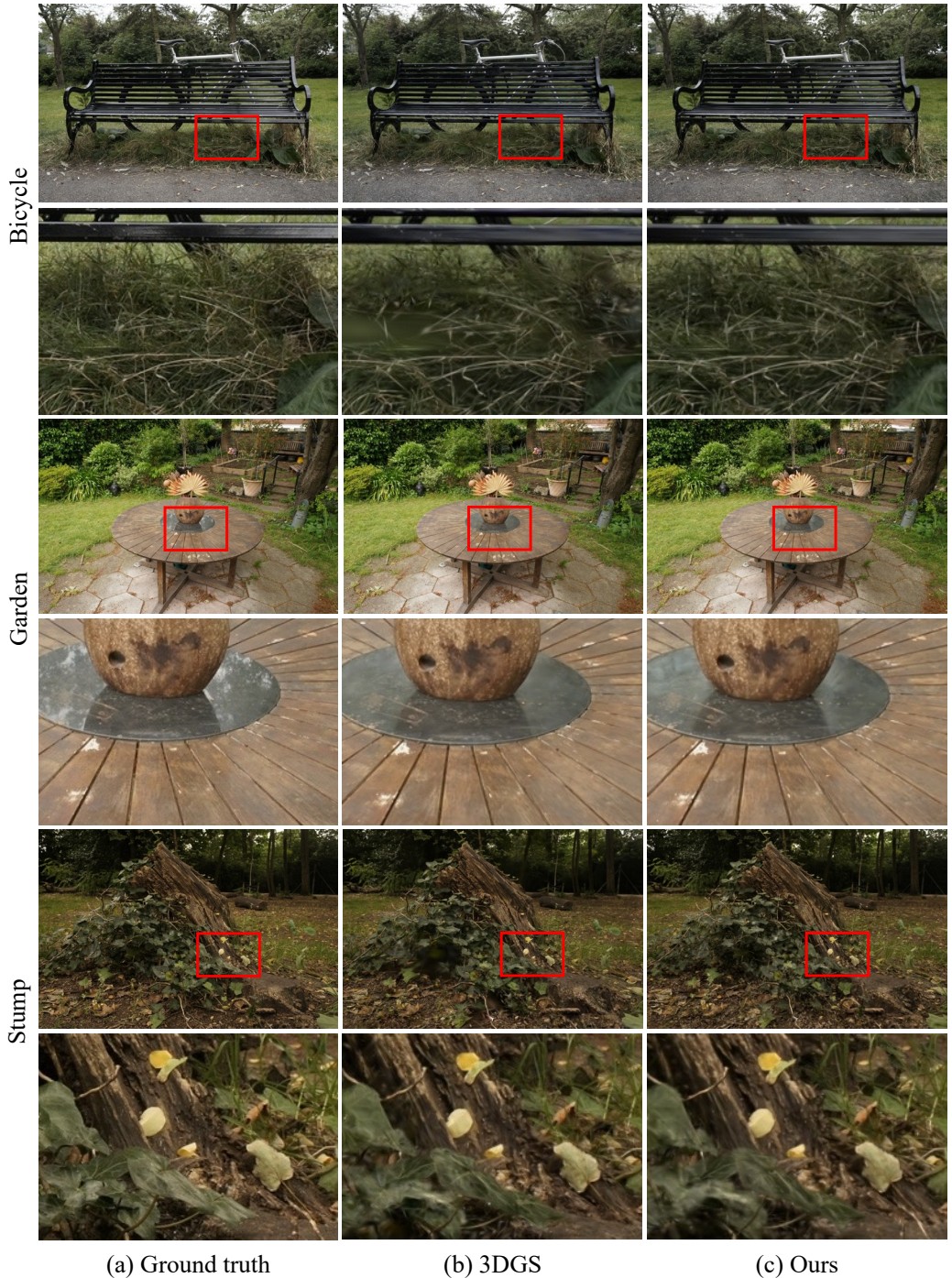

(a) Ground truth      (b) 3DGS      (c) Ours

Figure 11: Additional qualitative results on Mip-NeRF 360 (Barron et al., 2022).

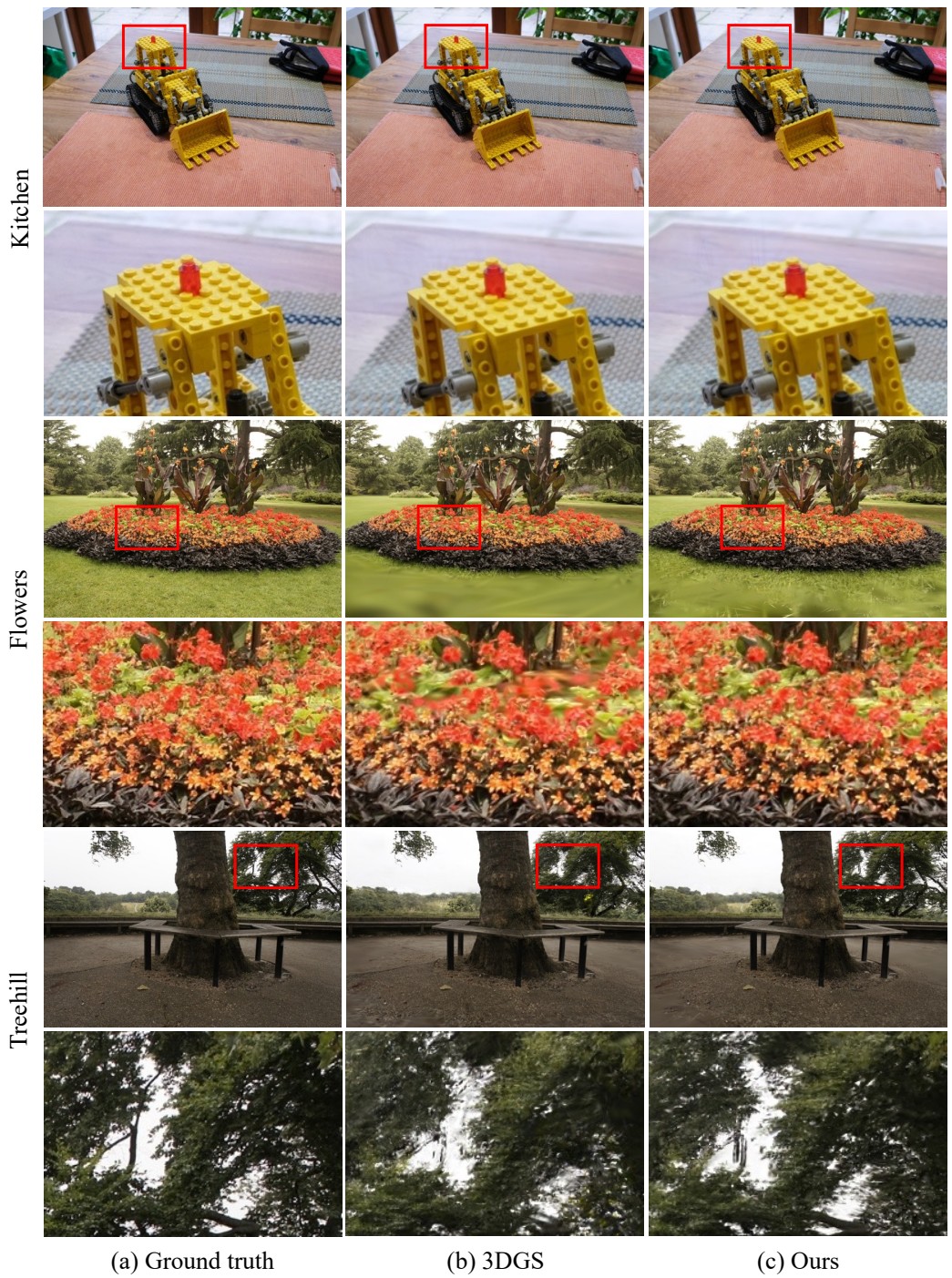

Figure 12: Additional qualitative results on Mip-NeRF 360 (Barron et al., 2022).

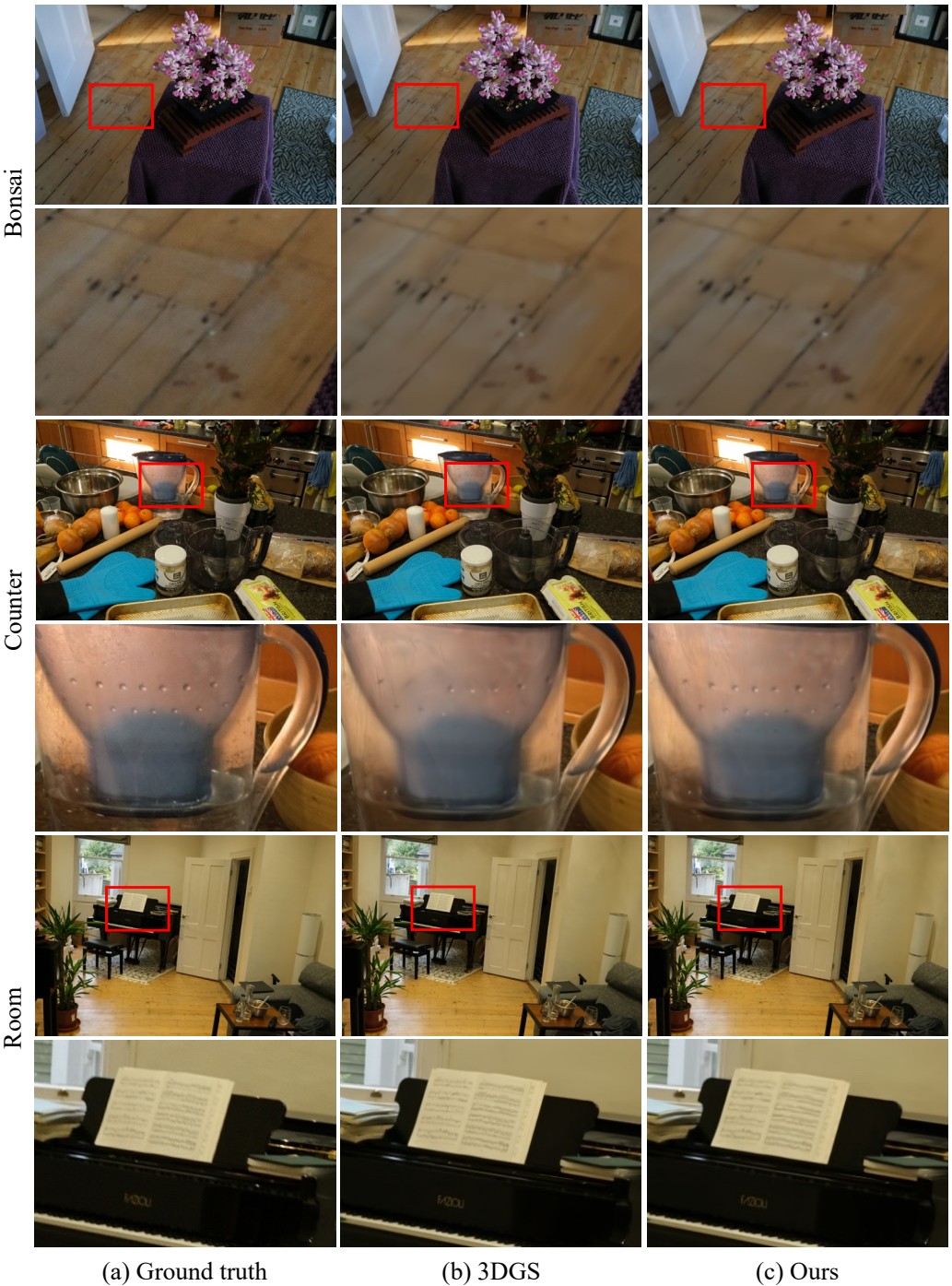

(a) Ground truth      (b) 3DGS      (c) Ours

Figure 13: Additional qualitative results on Mip-NeRF 360 (Barron et al., 2022).

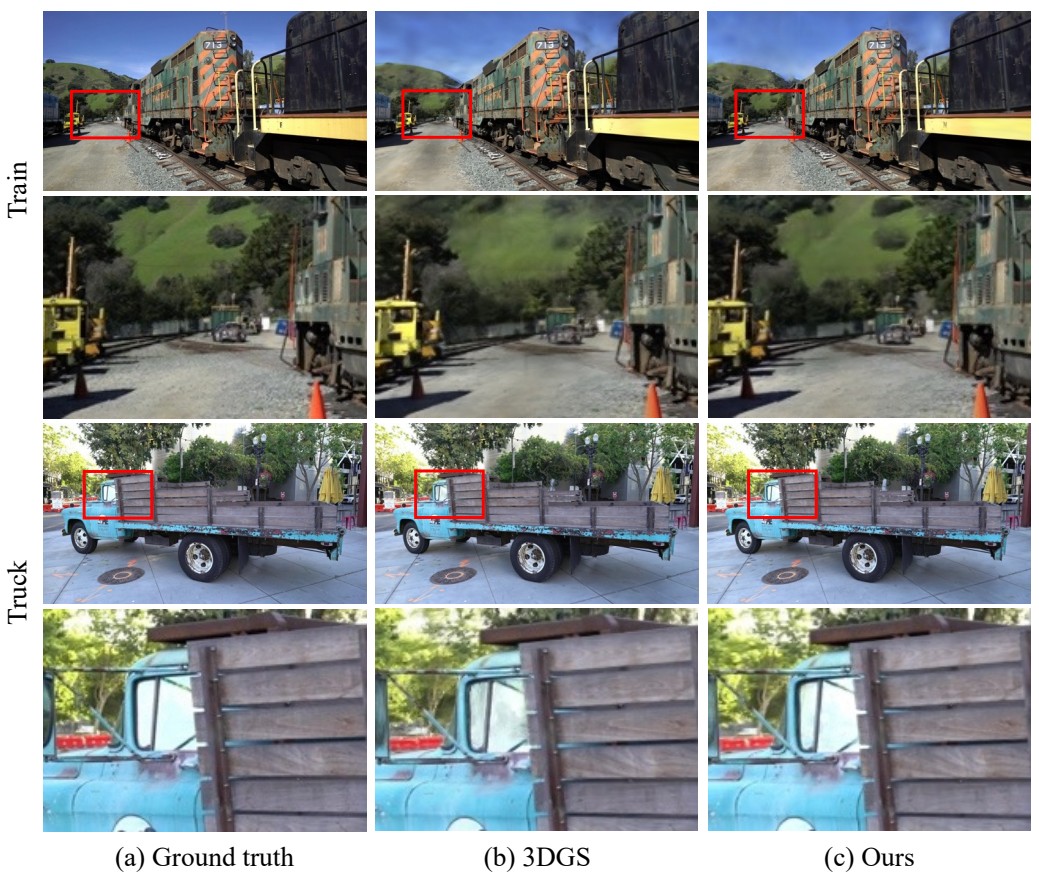

(a) Ground truth       (b) 3DGS       (c) Ours

Figure 14: Additional qualitative results on Tanks and Temples (Knapitsch et al., 2017).

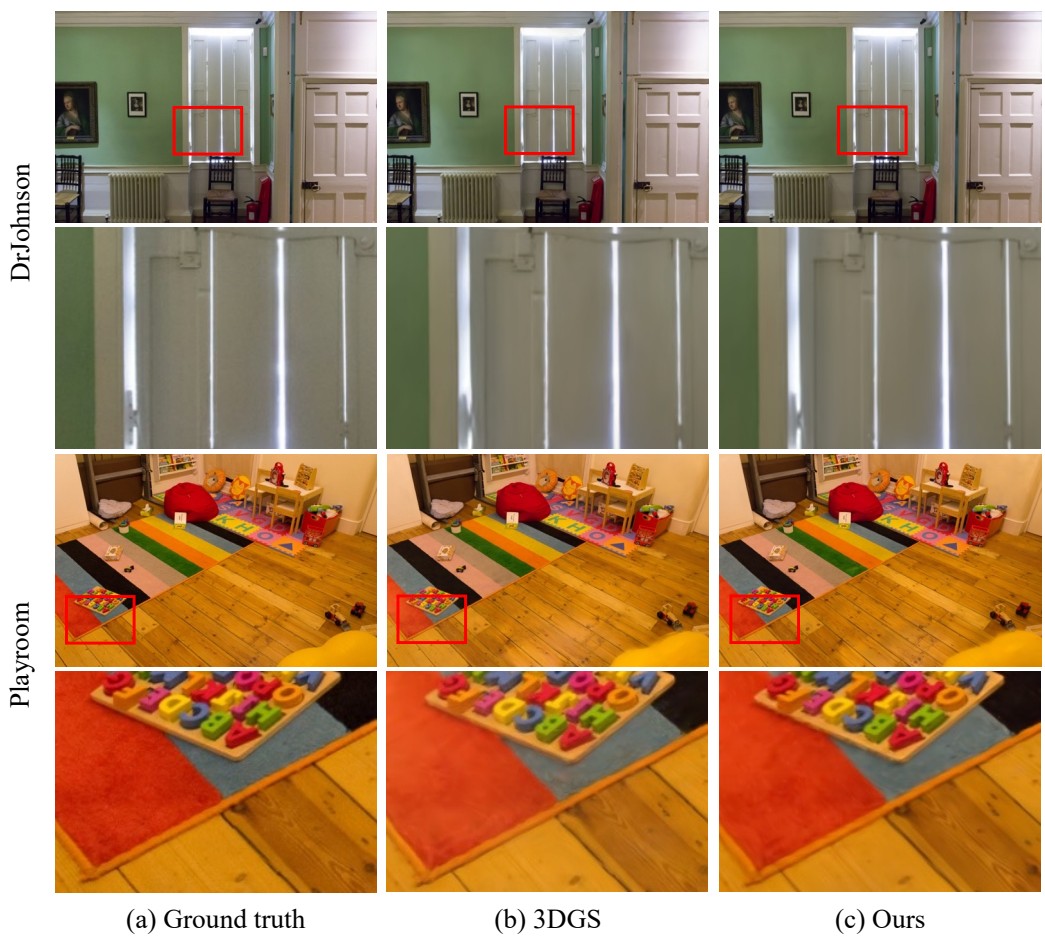

(a) Ground truth        (b) 3DGS        (c) Ours

Figure 15: Additional qualitative results on Deep Blending (Hedman et al., 2018).

