# OpenReview forum: "Locality-aware Gaussian Compression for Fast and High-quality Rendering"
_ICLR.cc/2025/Conference — ICLR 2025 Poster_

### Official Review · Reviewer_EsF7 · 2024-10-17

**Soundness:** 3
**Presentation:** 3
**Contribution:** 2
**Rating:** 3
**Confidence:** 5

**Summary:**

The paper introduces "LocoGS," a locality-aware 3D Gaussian Splatting (3DGS) framework designed for fast and high-quality rendering. This framework capitalizes on the spatial coherence among 3D Gaussians to offer a compact, efficient representation that significantly reduces storage requirements and enhances rendering speed without sacrificing quality.

Key contributions include:
1. A new locality-aware 3D Gaussian representation that efficiently encodes locally-coherent Gaussian attributes.
2. Implementation of additional components like dense initialization, Gaussian pruning, adaptive spherical harmonics bandwidth scheme, and tailored encoding schemes to maximize compression performance.
3. Demonstrated superior performance over existing methods in terms of compression ratio and rendering speed, verified through extensive experiments.

**Strengths:**

1. The locality-aware strategy for Gaussian attribute representation is a novel approach that leverages spatial coherence effectively.
2. The paper includes detailed comparisons with existing methods, showing improvements in storage efficiency and rendering speed. The reduced storage and increased speed facilitate the practical application of 3DGS in real-time scenarios, like mobile devices.
3. The motivation and clarity of the paper are commendable.

**Weaknesses:**

1. The multiple components introduced (e.g., multi-resolution hash grids, and adaptive SH bandwidths) might complicate the implementation and tuning of the framework.
2. The paper could benefit from more discussion on any potential trade-offs or limitations, particularly in different or more challenging rendering environments. e.g. metal and lighting area.

**Questions:**

1. Are there scenarios or conditions under which the proposed method might not perform as expected? e.g. large scenes.
2. Could you provide more detailed explanations or pseudo-codes for complex components like the adaptive SH bandwidth and pruning strategy? Are you going to publish your code?
3. Does your model also perform better in larger scenes? Can you please add more experiments on how it performs in larger scenes? e,g. Miller-19 dataset.

---

> ### Author Response · Authors · 2024-11-21
> **Response to Reviewer EsF7**
>
> We sincerely appreciate your insightful comments. We have tried to respond to all these valuable concerns as below. Please note that we have provided additional figures, tables, and videos.
>
> ---
>
> ### **[W1, Q2] Complicated implementation (plan to publish code)**
>
> We do not agree that our approach is particularly more complex than previous Gaussian compression approaches. Achieving high compression performance requires considering multiple properties of the 3D Gaussian. Therefore, previous approaches such as CompGS, Compact3DGS, EAGLES, LightGaussian, SSGS and HAC also consist of several steps, including several components such as pruning, quantization, sorting, hash-grids, and arithmetic encoding.
>
> Nevertheless, for better understanding, we provide pseudocode for the locality-aware Gaussian representation learning step of LocoGS, our main component, in Sec. A.4 in the revised appendix. We will also publicly release our source code as well as pre-trained checkpoints for benchmarking.
>
> ---
>
> ### **[W2, Q1] Potential trade-offs or limitations, (different or more challenging rendering environments. e.g. metal and lighting area)**
>
> As briefly discussed in the conclusion section in the manuscript, our method is not free from limitations. Beyond the training efficiency already mentioned in the submitted manuscript, our method has additional limitations. First, our Gaussian representation learning process requires more memory space than 3DGS due to the gradient computation of the hash grid and Gaussian attributes, which poses challenges for large-scale scenes. Further details on applying our method to large-scale scenes can be found in the global response.
>
> Second, our method assumes a cube-shaped, object-centric scene where target objects are located around the center of the scene. Based on this assumption, we set the hyperparameters of the hash grid, and adopt the contraction strategy of Mip-NeRF 360, which may be less effective for complex structures in large-scale scenes.
>
> Additionally, our method shares limitations with 3DGS, such as limited performance on specular surfaces, despite using a spherical harmonics-based color representation. Nevertheless, our method still performs similar to 3DGS even for such specular surfaces.
>
> We will include a more detailed discussion in the revision.
>
> ---
>
> ### **[Q3] Performance on large-scale datasets**
>
> Please refer to the [global response](https://openreview.net/forum?id=dHYwfV2KeP&noteId=wqrysnkItO).

---

> ### Author Response · Authors · 2024-11-25
>
> Dear Reviewer EsF7,
>
> Thank you for your insightful feedback. We have tried to address your concerns during the discussion period. Please let us know if you have any additional questions or concerns that need further discussion. We will promptly respond to your feedback.
>
> Best regards,
> Authors

---

> > ### Comment · Reviewer_EsF7 · 2024-11-26
> >
> > Thanks for the author's response.
> >
> > However, I have a few more comments:
> > 1. The metrics performance on large scenes is not good compared to 3DGS. Although it has 20% reduction in the number of Gaussian spheres and a good reduction in the storage size. When compared to the latest work on 3DGS compression, this article does not have too many notable highlights. However, it is precisely in large-scale scenarios where 3DGS truly needs to achieve compression and fast rendering, not the small scenes.
> >
> > 2. Currently, the novelty of this paper is not enough for ICLR standards compared to SOTA in 2024.
> >
> > 3. In Sec 4.1, the authors claim that the local coherence of Gaussian attributes is analogous to the pixel value coherence in natural images. While this analogy helps in understanding, it could benefit from a more detailed explanation or a mathematical backing to solidify the claim, especially on how exactly these properties translate between such different domains (Gaussians in 3D space versus pixel values in 2D images).
> >
> > 4. In Sec 5.2, the paper details different encoding schemes for various attributes but does not clearly compare these methods against potential alternatives or justify why these particular methods were chosen. Providing comparisons or rationale might strengthen this section, especially in terms of efficiency or error rates associated with each method.
> >
> > I will keep my rating.

---

> > > ### Author Response · Authors · 2024-11-28
> > > **Response to Reviewer EsF7 (1/2)**
> > >
> > > ### **[C1] Large-scale dataset**
> > >
> > > Please note that our method outperforms PSNR and LPIPS for the 'building' scene. This implies that our method achieves feasible rendering performance despite the large scale of the scene. Also, we believe that reducing the large storage size of scenes, such as the Mill-19 dataset, from over 1.3GB to 33MB is certainly critical for treating 3DGS in commonly used environment settings.
> > >
> > > In addition, handling large-scale scenes is an independent research area and is not addressed in existing 3DGS compression works. Managing large-scale scenes requires dedicated techniques, such as block-based processing, to overcome the physical limitations of GPU memory space, which is an active research area itself [1, 2]. Consequently, existing compression approaches have conducted their evaluations on datasets like Mip-NeRF 360, Tanks and Temples, and Deep Blending dataset, with some also adopting synthetic dataset and the BungeeNeRF dataset. Therefore, it is unfair to raise an issue on large-scale scenes only for our method.
> > >
> > > > 1. Lin et al., VastGaussian: Vast 3D Gaussians for Large Scene Reconstruction, CVPR 2024
> > > > 2. Liu et al., CityGaussian: Real-time High-quality Large-Scale Scene Rendering with Gaussians, ECCV 2024
> > >
> > > ---
> > >
> > > ### **[C2] Novelty**
> > >
> > > We respectfully disagree that the novelty of this paper is insufficient. This paper is the first to demonstrate the local coherence of all Gaussian attributes with comprehensive analysis while proposing an effective locality-exploiting framework to achieve both fast rendering speed and high compression rate. Please refer to the detailed differences in the responses to other reviewers ([response to reviewer 5DLr](https://openreview.net/forum?id=dHYwfV2KeP&noteId=5wB28EqIu6) and [response to Q1 of reviewer FMA7](https://openreview.net/forum?id=dHYwfV2KeP&noteId=2F8MnE4mwr)). Furthermore, our extensive experiments include all SOTA methods, even pre-printed arXiv papers, and demonstrate the superior performance of our method compared to them. We gently request the reviewer to provide any SOTA methods that were not included in our experiment.
> > >
> > > ---
> > >
> > > ### **[C3] Local coherence**
> > >
> > > We would like to note that we have provided a comprehensive analysis based on real data to validate the local coherence of Gaussians, and we believe that such an analysis on real data is the most straightforward way, and it is already sufficient to prove our claim. The analogy with a 2D image is only for helping better understanding, and we disagree that showing a mathematical connection between 3D Gaussians and 2D images is further needed for proving the local coherence of Gaussian attributes.
> > >
> > > Additionally, it is intuitive that spatially adjacent points in 3D space are likely to belong to the same object and thus have similar color and opacity. For example, consider two nearby points on a white solid door or a semi-transparent window. Regarding the shapes of Gaussians, visual examples from previous papers have shown that the shapes of Gaussians are influenced by texture or underlying geometry. For instance, the spokes of a bicycle wheel are often represented by elongated Gaussians, aligned with the spokes’ directions. These examples and our analysis demonstrate the local coherence of the shapes of Gaussians.

---

> > > ### Author Response · Authors · 2024-12-01
> > > **Kind Reminder**
> > >
> > > Dear Reviewer EsF7,
> > >
> > > As the discussion phase comes to a close, we gently remind you that we have tried to address all your additional concerns. If our responses have fully resolved the remaining issues, we would greatly appreciate it if you could acknowledge that in your rating. Please let us know if you have any additional questions or concerns that need further discussion. We will promptly respond to your feedback.
> > >
> > > Best regards,
> > > Authors

---

> ### Author Response · Authors · 2024-11-28
> **Response to Reviewer EsF7 (2/2)**
>
> ### **[C4] Encoding schemes**
>
> We would like to clarify that the encoding schemes in Sec. 5.2 are not our main contribution, and exploring all possible combinations of encoding schemes to find the optimal combination is beyond the scope of our paper. Instead, in our framework, we adopt commonly used compression techniques for each attribute following previous work. To compress positions, we apply octree encoding, which is the most common baseline compression method in point cloud compression approaches (e.g., D-PCC [3], SparsePCGC [4]). Quantization and entropy coding, employed for other attributes in our framework, are also one of the most basic compression techniques, which are also adopted in previous works (e.g., C3DGS, Compact-3DGS, and LightGaussian).
>
> > 3. He et al., Density-preserving Deep Point Cloud Compression, CVPR 2022
> > 4. Wang et al., Sparse Tensor-Based Multiscale Representation for Point Cloud Geometry Compression, TPAMI 2022
>
> We leverage Morton sorting to maintain the correspondence between the position and other attributes of each Gaussian, while it also helps compression as it places nearby Gaussians close to each other in the sorted sequence. In fact, any other sorting methods that work in a similar manner can be adopted. We found negligible differences in compression performance across different sorting methods, while all of them enhance the compression performance compared to random sorting in our preliminary experiments.
>
> We acknowledge that more optimal combinations of encoding schemes may exist. However, we stress that our main contribution lies in our locality-exploiting representation and the compression framework based on this representation. Seeking optimal combinations of encoding schemes is outside the scope of this paper.
>
> Nevertheless, as suggested by the reviewer, we provide additional details and analyses of encoding schemes. For quantization, we use k-bit uniform quantization due to its simplicity and computational efficiency. For compressing all other data, including base color, base scale, hash grid parameters, and MLP parameters, we considered widely used general-purpose lossless compression techniques such as Zip, GZip, and NPZ as our candidates. We found no notable difference between these methods regarding compression performance, so we chose NPZ, an entropy coding-based scheme, due to its convenience.
>
> Table R8 compares different compression schemes for different attributes. All the coding schemes in the table, except for G-PCC, are general-purpose lossless coding schemes, while G-PCC is an octree-based lossless coding scheme for point clouds. As the table shows, G-PCC outperforms all other general-purpose coding schemes for position encoding. For other data, all the general-purpose coding schemes perform similarly.
>
> **Table R8. Comparison of compression schemes. The numbers in the table are compressed sizes in MB.**
> ||Position (16-bit)|Base scale (6-bit)|Base color (8-bit)|Hash grid (6-bit)|MLP (16-bit)|
> |:---|:--:|:--:|:---:|:---:|:---:|
> |No compression|7.95|1.32|3.97|9.45|0.063|
> |Zip|5.97|0.80|3.11|6.83|0.056|
> |GZip|5.97|0.80|3.11|6.83|0.056|
> |NPZ|5.94|0.80|3.10|6.84|0.058|
> |G-PCC (Octree)|2.80|-|-|-|-|
>
>
> Table R9 compares the performance of *k*-bit uniform quantization and *k*-means clustering, a vector quantization method. While *k*-means clustering is a more sophisticated algorithm that can potentially result in higher compression performance, its performance severely relies on target data distribution. The table shows that *k*-bit uniform quantization performs better than *k*-means clustering for all data despite its simplicity, potentially due to the distributions being challenging to cluster.
>
> **Table R9. Comparison between *k*-means clustering and *k*-bit uniform quantization.**
> |Hash Grid|Base scale|Base color|PSNR&uarr;|SSIM&uarr;|LPIPS&darr;|Size (MB)&darr;|
> |:---|:---|:---|:--:|:--:|:---:|:---:|
> |*k*-means|*k*-means|*k*-means|27.07|0.815|0.217|15.88|
> |*k*-bit|*k*-means|*k*-means|27.08|0.815|0.217|14.82|
> |*k*-means|*k*-bit|*k*-means|27.36|0.816|0.217|15.84|
> |*k*-means|*k*-means|*k*-bit|27.06|0.814|0.218|15.34|
> |*k*-bit|*k*-bit|*k*-bit|27.33|0.814|0.219|13.89|
>
> We hope the details and analyses provided here help you understand our framework more clearly.

---

### Official Review · Reviewer_FMA7 · 2024-10-26

**Soundness:** 3
**Presentation:** 3
**Contribution:** 3
**Rating:** 6
**Confidence:** 4

**Summary:**

LocoGS analyzes the local coherence of the attributes of Gaussian primitives and introduces a novel representation that incorporates locality information. By utilizing the locality-aware 3D Gaussian representation along with other compression methods, such as quantization and encoding schemes, LocoGS achieves state-of-the-art compression performance and rendering FPS compared to existing compression methods.

**Strengths:**

1. LocoGS carefully analyzes the relationships among 3D Gaussian attributes and introduces a locality-aware representation based on these relationships.
2. The paper is well written and easy to follow, with an excellent categorization of compression methods.
3. LocoGS outperforms existing methods in both compression performance and rendering speed.

**Weaknesses:**

1. Dense initialization is not related to compression; it is merely a warm-up trick. It would be unfair for the authors to use this trick since all of the baselines still initialize with COLMAP points.
2. As we know, the training time for Nerfacto is significantly longer than for 3DGS. Why do the authors choose to use Nerfacto for warm-up instead of 3DGS? Both methods can generate coarse depth maps, which can then be used to create an initialization point cloud. Additionally, how many iterations do the authors train Nerfacto for dense initialization?
3. In the limitations section, the authors describe that LocoGS requires one hour more training time than the baselines. Which components affect the training efficiency—is it the dense initialization?
4. No demos submitted.

**Questions:**

1. I would like to understand the reason why the rendering performance of LocoGS is lower than that of Scaffold-GS. Aside from the quantization and encoding schemes, do the authors discard the view input of the MLP when obtaining each attribute of the 3D Gaussians?
2. How about the performance in large-scale datasets, like Mega-NeRF, Urbanscene3D or MatrixCity?

If the authors solve these questions, I’ll raise the score.

---

> ### Author Response · Authors · 2024-11-21
> **Response to Reviewer FMA7 (1/3)**
>
> We sincerely appreciate your insightful comments. We have tried to respond to all these valuable concerns as below. Please note that we have provided additional figures, tables, and videos.
>
> ---
>
> ### **[W1] Dense initialization is not related to compression. It would be unfair...**
>
> Although dense initialization is not directly related to compression, it is especially effective for compression when combined with compression techniques such as pruning and quantization. Thus, we proposed the dense initialization scheme as one component of our framework.
>
> [Fig. R2](https://drive.google.com/file/d/1_GVjCHH7k9d_TH3tFkTSyfv1vzxroGnt/view?usp=sharing) and [Table R3](https://openreview.net/forum?id=dHYwfV2KeP&noteId=WOPGP5zmOo) present comparisons against previous methods with dense initialization. Simply applying dense initialization to previous methods increases the number of Gaussians, leading to higher rendering quality, slower rendering speed, and larger storage size, as shown in Fig. R2(b) and Table R3. Fortunately, some compression techniques provide parameters for compression, such as pruning strength and quantization strength. By tuning such parameters, we can maintain the storage size even when using dense initialization. Fig. R2(c) and Table R3 show that dense initialization with compression parameter tuning can enhance the rendering quality while maintaining the rendering speed and storage size for most compression methods.
>
> More importantly, Fig. R2 and Table R3 show that our method still has distinctive advantages over existing methods with dense initialization and compression parameter tuning. Compared to them, our approach produces smaller-sized results while providing higher rendering speed and comparable rendering quality, thanks to our locality-aware approach. Please refer to the response for [W2 of reviewer 6k9A](https://openreview.net/forum?id=dHYwfV2KeP&noteId=WOPGP5zmOo) for more details. We will include the comparisons in the revision.
>
> **[[Link]](https://drive.google.com/file/d/1_GVjCHH7k9d_TH3tFkTSyfv1vzxroGnt/view?usp=sharing)**
> **Fig. R2. Evaluation of previous approaches applying dense initialization.**

---

> ### Author Response · Authors · 2024-11-21
> **Response to Reviewer FMA7 (2/3)**
>
> ### **[W2] Comparison of dense initialization methods (Nerfacto and 3DGS)**
>
> We use Nerfacto because Nerfacto requires a shorter training time than 3DGS for the same number of iterations and provides more accurate geometry information. Regarding the training time, Nerfacto reports that it takes 2 minutes for 5K iterations and 30 minutes for 70K iterations on a single A5000 GPU. On the other hand, Compact-3DGS reports that 3DGS takes 12 to 28 minutes for 30K iterations on a single A6000 GPU.
>
> For a clear comparison, we evaluated both methods in the same environment (RTX 3090). Table R6 reports the training times of Nerfacto and 3DGS on different scenes from the Mip-NeRF 360, Tanks and Temples, and Deep Blending datasets. We used 30K iterations for both methods, utilizing the publicly released codes with default settings. Note that 3DGS requires its own initialization, so we initialized it using COLMAP. On the other hand, Nerfacto does not require such an additional initialization and starts with random initialization. The table compares only the training times without initialization, and as shown, Nerfacto takes around half the training time of 3DGS.
>
> **Table R6. Evaluation of training time for 30K iterations.**
> ||Bonsai|Stump|Truck|Playroom|
> |:---|:---|:--:|:--:|:---:|
> |3DGS|20.0 min| 27.2 min| 17.8 min| 21.3 min|
> |Nerfacto|12.6 min|11.1 min|9.8 min|13.0 min|
>
>
> Additionally, we found that, while Nerfacto typically produces results with lower rendering quality than 3DGS, it generates more accurate geometry information thanks to the regularization terms in its training loss function, which helps our training process achieve more accurate Gaussian representation learning. Table R7 compares the final rendering qualities of LocoGS initialized with 3DGS and Nerfacto. As shown in the table, LocoGS with Nerfacto achieves higher rendering quality, faster rendering speeds, and fewer Gaussians, thanks to the better initial solutions provided by Nerfacto.
>
> **Table R7. Performance of LocoGS according to the initialization method.**
> ||PSNR&uarr;|SSIM&uarr;|LPIPS&darr;|Size (MB)&darr;|FPS&uarr;|#G|
> |:---|:---|:--:|:--:|:---:|:---:|:---:|
> |**Bonsai**|
> |3DGS|31.55|0.933|0.212|9.39|315|0.52 M|
> |Nerfacto|31.58|0.935|0.208|9.15|355|0.49 M|
> |**Stump**|
> |3DGS|27.04|0.790|0.199|17.63|281|2.02 M|
> |Nerfacto|27.24|0.801|0.194|15.55|303|1.65 M|
> |**Truck**|
> |3DGS|25.34|0.880|0.129|13.17|303|1.07 M|
> |Nerfacto|25.70|0.885|0.125|12.19|326|0.90 M|
> |**Playroom**|
> |3DGS|30.32|0.907|0.250|11.55|328|0.83 M|
> |Nerfacto|30.49|0.907|0.248|10.75|386|0.68 M|

---

> ### Author Response · Authors · 2024-11-21
> **Response to Reviewer FMA7 (3/3)**
>
> ### **[W3] Longer training time is a limitation.**
>
> Our neural field design and dense initialization result in a longer training time than 3DGS. This is primarily because the hash-grid-based neural field incurs additional computational costs compared to the direct optimization of attributes in 3DGS. Additionally, dense initialization produces a larger number of Gaussians than COLMAP at the beginning, leading to further computational overhead. To reduce the longer training time, adopting an effective culling strategy to pre-filter invisible Gaussians or employing a more efficient neural field architecture could be viable solutions.
>
> ---
>
> ### **[W4] No demos**
>
> We prepare example rendering videos and share them in the supplementary material. We will also release the code with pre-trained checkpoints once the paper gets accepted.
>
> ---
>
> ### **[Q1] Why lower rendering quality than Scaffold-GS**
>
> Scaffold-GS extends the original 3DGS framework by adopting MLPs for higher rendering quality. Specifically, Scaffold-GS decodes view-adaptive Gaussians using the view direction as input for MLPs. Thanks to the expressive power of MLPs, Scaffold-GS outperforms 3DGS in rendering quality. However, its per-view decoding process yields a large computational overhead for rendering each view. In our work, we do not follow the extension of Scaffold-GS to use the efficient rendering process of 3DGS. This may result in a minor rendering performance gap between Scaffold-GS and ours. However, as shown in Tables 1 and 2 of the manuscript, our method achieves comparable rendering quality while offering significantly faster rendering speeds.
>
> ---
>
> ### **[Q2] Performance on large-scale datasets**
>
> Please refer to the [global response](https://openreview.net/forum?id=dHYwfV2KeP&noteId=wqrysnkItO).

---

> > ### Comment · Reviewer_FMA7 · 2024-11-25
> >
> > After discussing with the author, all my doubts were resolved, and I will raise my score to 6.

---

### Official Review · Reviewer_5DLr · 2024-10-26

**Soundness:** 4
**Presentation:** 4
**Contribution:** 3
**Rating:** 8
**Confidence:** 4

**Summary:**

The paper aims to achieve both high rendering speed and reduced storage size for 3D Gaussian Splatting-based scene representations. To this end, it analyzes the local coherence of Gaussians across several scenes and proposes exploiting multi-resolution hash grids. More specifically, Gaussians now have only base scale, positions, and base color, while hash grids fill in the details. In addition, the paper employs adaptive SH bandwidth, pruning, dense initialization, quantization, and encoding. With all these methods, the paper demonstrates significant performance improvement.

**Strengths:**

The concept of locality is not novel in 3D Gaussian Splatting, but the paper effectively illustrates this through graphs. Moreover, it clearly addresses the differences between other locality-based methods (anchor-based methods), explaining why the proposed method is important and highlighting key points for readers already familiar with anchor-based representation.

The performance improvements in terms of size, rendering speed, and rendering quality are significant.

Additionally, the paper includes many technical details for adopting various techniques, especially during quantization and encoding, which will benefit those working in this research areas.

**Weaknesses:**

As I mentioned in the strength section, the idea of utilizing locality in 3D Gaussian Splatting is not novel.

**Questions:**

How did you calculate cosine similarity of opacities?
I would also like to know how and why three explicit attributes were selected.

---

> ### Author Response · Authors · 2024-11-21
> **Response to Reviewer 5DLr (1/2)**
>
> We sincerely appreciate your insightful comments. We have tried to respond to all these valuable concerns as below. Please note that we have provided additional figures, tables, and videos.
>
> ---
>
> ### **[Q1] Cosine similarity of opacities**
>
> As the opacity of each Gaussian is a scalar value, it is unsuitable for computing cosine similarity. Therefore, to evaluate the local coherence of the opacity values, we constructed vectors from the opacity values of nearby Gaussians and computed the cosine similarities between them. The other Gaussian attributes are vectors, so we directly computed their cosine similarities.
> We acknowledge that this approach may seem unintuitive and that cosine similarity may not be suitable for all Gaussian attributes. Consequently, we plan to replace cosine similarity with Euclidean distance in the revision.
> [Fig. R3](https://drive.google.com/file/d/1yhzhSbuNvLMdHkNOhBiRECfLI_Cl7C_u/view?usp=sharing) presents analyses using the Euclidean distance. The top row in [Fig. R3](https://drive.google.com/file/d/1yhzhSbuNvLMdHkNOhBiRECfLI_Cl7C_u/view?usp=sharing) shows histograms of the Euclidean distances between two Gaussians for each attribute. Following our submitted manuscript, the pink histograms represent spatially adjacent Gaussian pairs, while the yellow histograms represent spatially distant Gaussian pairs. Regardless of the spatial distances, all histograms have high peaks at small Euclidean distances, while their shapes become flatter as the spatial distances between Gaussians increase. The bottom row in [Fig. R3](https://drive.google.com/file/d/1yhzhSbuNvLMdHkNOhBiRECfLI_Cl7C_u/view?usp=sharing) shows the average distances between two Gaussians for each attribute. Similarly, the pink bars represent spatially adjacent Gaussians, and the yellow bars represent spatially distant Gaussians. The graphs demonstrate that spatially adjacent Gaussians have smaller Euclidean distances between their attributes. These analyses clearly indicate the local coherence of the Gaussian attributes.
>
> **[[Link]](https://drive.google.com/file/d/1yhzhSbuNvLMdHkNOhBiRECfLI_Cl7C_u/view?usp=sharing)**
> **Fig. R3. Evaluation of the local coherence of Gaussian attributes. We measure the Euclidean distance of neighboring Gaussian attributes.**

---

> ### Author Response · Authors · 2024-11-21
> **Response to Reviewer 5DLr (2/2)**
>
> ### **[Q2] Design of explicit attributes**
>
> The explicit attributes are selected based on the purpose of optimization as described in L229-L233, not based on their local coherence. The optimization process of 3DGS, which involves explicit initialization of the Gaussian attributes and the adaptive growing strategy that directly adjusts the scale and base color of each Gaussian primitive, necessitates some of the Gaussian attributes to be stored in an explicit form for each Gaussian. Therefore, we define the explicit attributes as the minimal set of attributes that need to be stored in an explicit form (position, base scale, and base color) while defining the implicit attributes as the remaining attributes that can be stored in a compact neural field.
>
> To validate the optimality of our current selection of explicit attributes, we compared the performance of LocoGS when the base color and base scale are treated as either explicit or implicit attributes. Table R5 summarizes the comparison results. As the table shows, treating either the base scale or base color as implicit attributes degrades performance, as they cannot be explicitly initialized or manipulated during the training process. Specifically, treating the base scale as an implicit attribute makes the training process highly unstable, resulting in a severely small number of Gaussians and poor rendering quality. Treating the base color as an implicit attribute also degrades the rendering quality while increasing the number of Gaussians. Thus, we selected them as explicit attributes along with positions.
>
> **Table R4. Ablation study on the attribute design.**
> |Dataset|Color|Scale|PSNR&uarr;|SSIM&uarr;|LPIPS&darr;|Size (MB)&darr;|FPS&uarr;|#G|
> |:---|:---|:---|:--:|:--:|:---:|:---:|:---:|:---:|
> |***Mip-NeRF 360***|Explicit|Implicit|20.47|0.661|0.383|10.13|176|0.65 M|
> ||Implicit|Explicit|27.15|0.811|0.225|11.22|224|1.42 M|
> ||Explicit|Explicit|27.33|0.814|0.219|13.89|270|1.32 M|
> ||
> |***Tanks and Temples***|Explicit|Implicit|17.32|0.660|0.391|11.76|224|0.86 M|
> ||Implicit|Explicit|23.71|0.849|0.166|10.88|271|1.09 M|
> ||Explicit|Explicit|23.84|0.852|0.161|12.34|311|0.86 M|

---

> > ### Comment · Reviewer_5DLr · 2024-11-23
> >
> > I appreciate the authors for providing additional results to address the concerns raised by the reviewers.
> >
> > While the paper could be beneficial for those working on compressing 3D Gaussian Splatting (3DGS), I still have concerns about the novelty of exploiting locality. The related works section currently focuses mainly on anchor-based methods' aspects like quality and storage but does not adequately address the locality-related properties of them.
> >
> > To strengthen the paper, it would be helpful to clearly highlight the differences between the proposed method and locality-exploiting methods more properly.

---

> ### Author Response · Authors · 2024-11-24
> **Response to Reviewer 5DLr**
>
> We appreciate the reviewer’s feedback. To highlight the novelty and effectiveness of our locality-aware strategy, we compare our method with previous approaches, emphasizing the differences in locality utilization below. We will revise the paper to make these differences more apparent.
>
> We may categorize existing 3D Gaussian compression approaches into global-statistic-based approaches, and locality-exploiting approaches. Global-statistic-based approaches include quantization-based approaches (CompGS, LightGaussian, and EAGLES), and an image-codec-based approach (SSGS). These approaches do not exploit the local coherence of Gaussian attributes, but they rely on global statistics of Gaussian attributes for quantization, and for image-codec-based compression.
>
> Locality-exploiting approaches include C3DGS, Compact-3DGS, and anchor-based approaches. However, these approaches restrictively exploit locality and achieve limited performance compared to ours.
>
> Specifically, C3DGS sorts Gaussian attributes according to their positions along a space-filling curve. This sorting process places spatially close Gaussians in the 3D space at nearby positions in a sorted sequence, enabling subsequent compression steps to exploit the local coherence of Gaussians. However, projecting 3D Gaussians to a 1D sequence along a space-filling curve cannot fully leverage the local coherence of Gaussian attributes. In contrast, our method directly encodes local information in an effective structure, a multi-resolution hash grid, during optimization. Therefore, our framework can more effectively exploit local coherence in the 3D space.
>
> Compact-3DGS leverages a hash grid to encode view-dependent colors. However, for geometric attributes including rotation and scale, it uses vector quantization, which exploits the global statistics instead of local coherence. As a result, while Compact-3DGS achieves noticeable compression performance with an 81.8x compression ratio for color, it achieves only a 5.5x compression ratio for vector-quantized attributes. In contrast, we reveal that geometric attributes also exhibit local coherence in our analysis in Sec. 4.1. Based on this analysis, we propose a locality-aware framework that exploits local coherence not only for view-dependent color but also for geometric attributes, achieving superior compression performance.
>
> Anchor-based methods like Scaffold-GS and HAC also exploit local coherence for Gaussian attributes. Scaffold-GS uses an anchor-point-based representation for high-quality rendering, where each anchor point encodes multiple locally-adjacent Gaussians with attributes such as shapes and colors changing based on the viewing direction. However, Scaffold-GS focuses on rendering quality rather than compression and requires computationally expensive per-view processing. HAC builds on Scaffold-GS, incorporating a hash grid for compression. However, unlike our method, HAC encodes only the context of anchor point features using a hash grid. Also, despite its high rendering quality and compression ratio, HAC requires slow per-view rendering due to its anchor-point-based representation.

---

> > ### Comment · Reviewer_5DLr · 2024-11-25
> >
> > I appreciate the author's prompt and detailed explanation.
> > Thank you.

---

### Official Review · Reviewer_6k9A · 2024-11-02

**Soundness:** 3
**Presentation:** 3
**Contribution:** 3
**Rating:** 6
**Confidence:** 5

**Summary:**

This paper tackles the high storage demands associated with 3D Gaussian Splatting by introducing an effective compression method. The core idea is to exploit local similarity within Gaussian attributes, representing them through compact, well-encoded local features. The authors start by conducting a statistical analysis of 3D Gaussian Splatting (3DGS) attributes and proceed to design a local feature that captures and encodes these similar patterns within a given region. By combining carefully structured steps—such as pruning, point cloud initialization, quantization, and entropy encoding—the proposed method achieves substantial compression of the 3DGS field. Remarkably, this compression is achieved while maintaining, or even enhancing, the rendering quality and fidelity relative to the original setup.

**Strengths:**

1. The overall solution is comprehensive, incorporating careful designs for initialization, pruning, and compression schemes for different components.
2. The paper is well-written with a clear logical flow.
3. The dense initialization demonstrates an interesting improvement in compression.
4. The analysis of storage size in Tab.3 provides a valuable indication for the community about the current bottlenecks in compression.

**Weaknesses:**

1. Number of Gaussians in Variants: It would be beneficial to provide the number of Gaussians for different variants, as Ours-Sparse has a lower FPS compared to Ours-Small and Ours. Understanding whether the final number of Gaussians influenced the results is important. Including the point number counts along the training iteration would help illustrate the influence of different initializations. Additionally, it would be interesting to see if such an initialization design could improve the performance of previous approaches.
2. The method still takes extra training time to compress a 3DGS, which will be a limitation in some real applications.

**Questions:**

1. Please include the number of Gaussians for the different ablation versions in Tab.3 to understand the main reason for high FPS.

2 Decoding Time: Including the decoding time would help readers understand the applicability of the proposed method.

---

> ### Author Response · Authors · 2024-11-21
> **Response to Reviewer 6k9A (1/3)**
>
> We sincerely appreciate your insightful comments. We have tried to respond to all these valuable concerns as below. Please note that we have provided additional figures, tables, and videos.
>
> ---
>
> ### **[W1] Number of Gaussians**
>
> We report the number of Gaussians of each variant in Table R2. Compared to 'Ours' and 'Ours-Sparse (Ours w/o dense initialization)', 'Ours-Small' has fewer Gaussians, resulting in higher rendering speed and slight degradation in rendering quality. In our framework, the rendering speed is inversely proportional to the number of Gaussians. While the rendering quality is influenced by the number of Gaussians, as shown in the table, it is also affected by other factors, such as dense initialization.
>
> **Table R2. Evaluation of rendeing quality, storage size, rendering speed, and number of Gaussians.**
> |Method|PSNR&uarr;|SSIM&uarr;|LPIPS&darr;|Size (MB)&darr;|FPS&uarr;|#G|
> |:---|:--:|:--:|:---:|:---:|:---:|:---:|
> |***Mip-NeRF 360***||
> |Our (w/o dense init.)|27.16|0.807|0.233|14.15|246|**1.38 M**|
> |Ours-Small|27.04|0.806|0.232|7.90|310|**1.09 M**|
> |Ours|27.33|0.814|0.219|13.89|270|**1.32 M**|
> |***Tanks and Temples***||
> |Our (w/o dense init.)|23.81|0.846|0.183|12.28|301|**0.87 M**|
> |Ours-Small|23.63|0.847|0.169|6.59|333|**0.78M**|
> |Ours|23.84|0.852|0.161|12.34|311|**0.89M**|
> |***Deep Blending***||
> |Our (w/o dense init.)|30.01|0.906|0.249|13.82|257|**1.28 M**|
> |Ours-Small|30.06|0.904|0.249|7.64|334|**1.04 M**|
> |Ours|30.11|0.906|0.243|13.38|297|**1.20 M**|
>
>
>
>
> We also provide the numbers of Gaussians along the training iterations in [Fig. R1](https://drive.google.com/file/d/1OPxx2vby48BDNmhJywnmDobv83pItKNH/view?usp=sharing). In the figure, 'Ours' and 'Ours-small' begin with dense initialization. Thus, at the beginning, they have more Gaussians than 3DGS and 'Ours (w/o dense init.)'. We prune Gaussians for the entire optimization process (30K), whereas 3DGS preserves a large number of Gaussians after densification (15K). Therefore, we can achieve a small Gaussian quantity after optimization. 'Ours-Small' uses a larger weight for the mask loss, so it has fewer Gaussians than 'Ours' after training.
>
> **[[Link]](https://drive.google.com/file/d/1OPxx2vby48BDNmhJywnmDobv83pItKNH/view?usp=sharing)**
> **Fig. R1. Number of Gaussians along iterations.**

---

> ### Author Response · Authors · 2024-11-21
> **Response to Reviewer 6k9A (2/3)**
>
> ### **[W2] Dense initialization for previous approaches**
>
> Simply applying dense initialization to previous methods results in an increase in the number of Gaussians, leading to higher rendering quality. However, this also causes an increase in storage size and a decrease in rendering speed. To avoid the increase in the number of Gaussians, we can adjust the compression parameters of the existing compression methods, such as pruning strength and quantization strength. When properly combined with such parameter tuning, dense initialization can still enhance the performance of these existing compression methods regarding rendering speed, storage requirement, and rendering quality since dense initialization provides a better initial solution than COLMAP-based initialization.
>
> To validate this, we conducted an additional evaluation comparing the performance of existing approaches without dense initialization, with dense initialization but no compression parameter tuning, and with dense initialization and compression parameter tuning on the MipNeRF-360 dataset. [Fig. R2](https://drive.google.com/file/d/1_GVjCHH7k9d_TH3tFkTSyfv1vzxroGnt/view?usp=sharing) and Table R3 summarize the evaluation results. In this evaluation, we also included two non-compression methods, 3DGS and Scaffold-GS, which have no compression parameters, so we did not perform compression parameter tuning for them. As shown in the figure and table, dense initialization without compression parameter tuning increases the number of Gaussians and storage size, and it decreases the rendering speed for all methods. On the other hand, dense initialization with compression parameter tuning enhances rendering speed and storage requirements while achieving comparable or higher rendering quality for most existing approaches. It is worth mentioning that dense initialization with compression parameter tuning does not perform well with HAC, as shown in Table R3, because its anchor-based approach does not directly encode individual Gaussians, thus lacks a pruning mechanism to reduce the number of Gaussians.
>
> Finally, [Fig. R2(c)](https://drive.google.com/file/d/1_GVjCHH7k9d_TH3tFkTSyfv1vzxroGnt/view?usp=sharing) shows that our approach still has distinctive advantages over existing compression methods with dense initialization and compression parameter tuning. Compared to them, our approach produces smaller-sized results while providing higher rendering speed and comparable rendering quality. This result again proves the effectiveness of our locality-aware approach.
>
> **[[Link]](https://drive.google.com/file/d/1_GVjCHH7k9d_TH3tFkTSyfv1vzxroGnt/view?usp=sharing)**
> **Fig. R2. Evaluation of previous approaches applying dense initialization.**
>
> **Table R3. Ablation study on dense initialization. * denotes applying dense initialization and † denotes applying dense initialization with stronger compression parameters.**
> |Method|PSNR&uarr;|SSIM&uarr;|LPIPS&darr;|Size (MB)&darr;|FPS&uarr;||Method|PSNR&uarr;|SSIM&uarr;|LPIPS&darr;|Size (MB)&darr;|FPS&uarr;|
> |:---|:---|:--:|:--:|:---:|:---:|:---|:---|:--:|:--:|:---:|:---:|:---:|
> |__3DGS__|27.44|0.813|0.218|822.6|127||__LightGS__|26.90|0.800|0.240|53.96|244|
> |__3DGS*__|27.80|0.826|0.198|935.8|118||__LightGS*__|27.08|0.806|0.229|60.21|228|
> |__Scaffold-GS__|27.66|0.812|0.223|187.3|122||__LightGS†__|26.86|0.799|0.241|49.64|247|
> |__Scaffold-GS*__|28.24|0.828|0.187|387.4|49||__EAGLES__|27.10|0.807|0.234|59.49|155|
> |__C3DGS__|27.09|0.802|0.237|29.98|134||__EAGLES*__|27.49|0.821|0.214|67.54|126|
> |__C3DGS*__|27.39|0.812|0.220|33.87|131||__EAGLES†__|27.35|0.818|0.219|56.55|137|
> |__C3DGS†__|27.39|0.812|0.221|26.95|141||__SSGS__|27.02|0.800|0.226|43.77|134|
> |__Compact3DGS__|26.95|0.797|0.244|26.31|143||__SSGS*__|27.13|0.805|0.211|72.00|102|
> |__Compact3DGS*__|27.47|0.814|0.219|31.99|124||__SSGS†__|26.84 |0.797|0.216|39.10|102|
> |__Compact3DGS†__|27.30|0.807|0.232|20.19|188||__HAC__|27.49|0.807|0.236|16.95|110|
> |__CompGS__|27.04|0.804|0.243|22.93|236||__HAC*__|27.71|0.822|0.201|42.67|46|
> |__CompGS*__|27.42|0.818|0.223|24.75|218||__HAC†__|27.16|0.808|0.219|32.19|54|
> |__CompGS†__|27.21|0.810|0.241|15.24|271||__Ours__|27.33|0.814|0.219|13.89|270|

---

> ### Author Response · Authors · 2024-11-21
> **Response to Reviewer 6k9A (3/3)**
>
> ### **[Q1] Ablation on the number of Gaussians**
>
> Thank you for your thoughtful feedback. We have added the number of Gaussians in Table 3 in the revised manuscript.
>
> ---
>
> ###  **[Q2] Decoding time**
> Table R4 compares the decoding times of different compression methods. LocoGS shows comparable decoding times despite its highly compact storage sizes compared to the other approaches, requiring less than 9 seconds for a single scene.
>
> **Table R4. Evaluation of the decoding time (sec).**
> ||Mip-NeRF 360|Tanks and Temples|Deep Blending|
> |:---|:--:|:--:|:---:|
> |CompGS|12.9|6.9|10.6|
> |Compact3DGS|178.4|139.4|59.8|
> |C3DGS|0.2|0.1|0.2|
> |EAGLES|8.1|4.1|7.0|
> |LightGaussian|0.7|0.5|0.8|
> |SSGS|6.6|3.8|2.8|
> |HAC|24.3|13.9|5.0|
> |Ours-Small|7.3|4.9|6.7|
> |Ours|8.5|5.5|7.9|

---

> ### Author Response · Authors · 2024-11-25
>
> Dear Reviewer 6k9A,
>
> Thank you for your insightful feedback. We have tried to address your concerns during the discussion period. Please let us know if you have any additional questions or concerns that need further discussion. We will promptly respond to your feedback.
>
> Best regards,
> Authors

---

> > ### Comment · Reviewer_6k9A · 2024-11-26
> > **Response**
> >
> > Thanks for the author's response. I highly suggest including these experiments and analyses in the revised version. My concerns are almost solved. I will maintain my rating.

---

### Author Response · Authors · 2024-11-21
**Global Response by Authors**

## **Global response**

Dear reviewers,

We thank all reviewers for their thoughtful comments. We have tried to respond to all these valuable concerns. In response to the comments, we have revised the manuscript and provided additional supplementary materials. Please note that we have highlighted updates in blue.

---

### **[Q1] Performance on large-scale datasets (FMA7, EsF7)**

As requested by the reviewers, we evaluated the performance of LocoGS on a representative large-scale dataset, the Mill-19 dataset, as shown in Table R1. The large-scale scenes in Mill-19 require an excessive number of Gaussians, resulting in large storage requirements (>1.3GB) for the original 3DGS approach. LocoGS successfully reduces the memory size to less than 34 MB, achieving a **47.2x compression ratio**. Additionally, our effective compression scheme involving Gaussian point pruning also substantially improves the rendering speed.

Nevertheless, despite its high compression ratio, our method requires approximately 1.5 to 2 times more memory space than the original 3DGS during training due to the gradient computation of the hash grid and Gaussian attributes. This increased memory requirement presents some challenges when applying our method directly to large-scale scenes. The slight quality degradation observed in Table R1 is also because of this heavy memory requirement, which limits the number of Gaussians to fewer than necessary. To address this issue, dividing a large-scale scene into smaller subregions could be a promising solution. By breaking the scene into more manageable parts, we can reduce the memory requirement and maintain high rendering quality across the entire scene. Please note that increased memory consumption only applies to the training phase. Our method follows memory usage of the original 3DGS for the rendering, which is proportional to the number of Gaussians.

**Table R1. Evaluation on the Mill-19 dataset.**
|Method|PSNR&uarr;|SSIM&uarr;|LPIPS&darr;|Size (MB)&darr;|FPS&uarr;|#G|
|:---|:--:|:--:|:---:|:---:|:---:|:---:|
|***Building***||
|3DGS|19.14|0.651|0.365|1371.02|83|5.44 M|
|Ours|19.79|0.630|0.354|32.97|113|4.13 M|
|***Rubble***|||
|3DGS|23.75|0.733|0.311|1752.37|69|6.95 M|
|Ours|23.38|0.651|0.366|33.26|122|4.20 M|

---

### Author Response · Authors · 2024-11-28
**Revised Manuscript**

Dear reviewers and AC,

We sincerely appreciate the reviewers’ insightful comments on how to enhance our paper. To address these valuable suggestions, we have carefully revised the manuscript as follows:

**[Section 2]**
Locality-exploiting 3DGS approaches (5DLr)

**[Section 4.1 & Figure 1]**
Analysis of local coherence based on Euclidean distance (5DLr)

**[Table 3]**
Number of Gaussians (6k9A)

**[Section 7]**
Limitations for large-scale scenes (FMA7, EsF7)

**[Section A.3.1]**
Ablation study on the attribute design (5DLr)

**[Section A.3.2]**
Ablation study on the hash grid size. We relocated this section from Sec. 6.2 due to the page limit.

**[Section A.3.4]**
Effect of dense initialization on existing compression methods (6k9A, FMA7)

**[Section A.3.5]**
Ablation study on the dense initialization method (FMA7)

**[Section A.4]**
Algorithms for locality-aware Gaussian representation learning (EsF7)

**[Section A.5.1]**
Number of Gaussians (6k9A)

**[Section A.5.2]**
Decoding time (6k9A)

**[Section A.5.3]**
Large-scale Scenes (FMA7, EsF7)

Please note that these updates have been highlighted in blue.

Best regards,
Authors

---

### Meta-Review · Area_Chair_prcP · 2024-12-21

**Metareview:**

This paper introduces a method for 3D Gaussian compression by leveraging the spatial coherence of 3D Gaussians. The results demonstrate that the method effectively reduces storage requirements and improves rendering speed. While several concerns were initially raised, most were addressed in the rebuttal. The remaining issues are related to the method's novelty and its applicability to large-scale scenes.

In terms of novelty, while similar concepts have been explored, the rebuttal clarified the distinctions between the proposed approach and other locality-exploiting methods. Regarding large-scale scenes, the rebuttal demonstrated that the proposed method effectively reduces the storage size of a scene from 1.3GB to 33MB. While the compression method is more beneficial for larger scenes, as stated in the rebuttal, compressing extremely large scenes would require additional effort in scene and memory management, which goes beyond the scope of this paper. Additionally, it is worth noting that most papers on the same topic conduct experiments on similar datasets.

**Additional Comments On Reviewer Discussion:**

The reviews identified several issues, including performance on large-scale datasets, the number of Gaussians, dense initialization, decoding time, the design of explicit attributes, implementation complexity, and limitations. The rebuttal effectively addressed most of these concerns.

The remaining major issues primarily relate to the method's novelty and its applicability to large-scale scenes. While similar concepts have been explored, the rebuttal clarified the distinctions between this approach and existing methods, with most reviewers agreeing on the clarification. Additionally, compressing extremely large scenes is beyond the scope of this paper, and the paper's evaluation aligns with the standards set by most papers on the same topic.

---

### Decision · Program_Chairs · 2025-01-22

Accept (Poster)